# Integrative genotyping of cancer and immune phenotypes by long-read sequencing

Livius Penter [1,2,3,4,5,9], Mehdi Borji[1,2,6,9], Adi Nagler[1,2,9], Haoxiang Lyu [6], Wesley S. Lu[6], Nicoletta Cieri [1,2,3], Katie Maurer [1,2,3], Giacomo Oliveira [1,2,3], Aziz M. Al'Khafaji [2], Kiran V. Garimella[2], Shuqiang Li [1,2,6], Donna S. Neuberg [7], Jerome Ritz [1,3,8], Robert J. Soiffer[1,3,8], Jacqueline S. Garcia[1,3], Kenneth J. Livak [1,6,10] & Catherine J. Wu[1,2,3,8,10] ✉

Single-cell transcriptomics has become the definitive method for classifying cell types and states, and can be augmented with genotype information to improve cell lineage identification. Due to constraints of short-read sequencing, current methods to detect natural genetic barcodes often require cumbersome primer panels and early commitment to targets. Here we devise a flexible long-read sequencing workflow and analysis pipeline, termed *nanoranger*, that starts from intermediate single-cell cDNA libraries to detect cell lineage-defining features, including single-nucleotide variants, fusion genes, isoforms, sequences of chimeric antigen and TCRs. Through systematic analysis of these classes of natural 'barcodes', we define the optimal targets for nanoranger, namely those loci close to the 5' end of highly expressed genes with transcript lengths shorter than 4 kB. As proof-of-concept, we apply nanoranger to longitudinal tracking of subclones of acute myeloid leukemia (AML) and describe the heterogeneous isoform landscape of thousands of marrow-infiltrating immune cells. We propose that enhanced cellular genotyping using *nanoranger* can improve the tracking of single-cell tumor and immune cell co-evolution.

Single cell RNA sequencing (scRNA-seq) has revealed the remarkable heterogeneity within cellular identities of cancer and is increasingly used for longitudinal tracking of cell states to understand how therapeutic interventions reshape gene expression profiles[1,2]. A crucial prerequisite of cancer single cell studies is the unambiguous identification of malignant versus physiologic populations, oftentimes impossible based on gene expression profiles alone. Multi-omics approaches that combine different single cell assays can be used to overcome this challenge, but the integration of independently acquired data layers remains typically complex.

[1]Department of Medical Oncology, Dana-Farber Cancer Institute, Boston, MA, USA. [2]Broad Institute of Massachusetts Institute of Technology and Harvard University, Cambridge, MA, USA. [3]Harvard Medical School, Boston, MA, USA. [4]Department of Hematology, Oncology, and Tumorimmunology, Campus Virchow Klinikum, Berlin, Charité - Universitätsmedizin Berlin, corporate member of Freie Universität Berlin and Humboldt-Universität zu Berlin, Berlin, Germany. [5]Berlin Institute of Health at Charité – Universitätsmedizin Berlin, BIH Biomedical Innovation Academy, BIH Charité Digital Clinician Scientist Program, Charitéplatz 1, 10117 Berlin, Germany. [6]Translational Immunogenomics Lab, Dana-Farber Cancer Institute, Boston, MA, USA. [7]Department of Data Science, Dana-Farber Cancer Institute, Boston, MA, USA. [8]Department of Medicine, Brigham and Women's Hospital, Boston, MA, USA. [9]These authors contributed equally: Livius Penter, Mehdi Borji, Adi Nagler. [10]These authors jointly supervised this work: Kenneth J. Livak, Catherine J. Wu. ✉ e-mail: catherine_wu@dfci.harvard.edu

To address this problem, natural barcodes such as copy number variants (CNV), somatic nuclear and mitochondrial DNA (mtDNA) mutations are increasingly used to aid in the identification of malignant clones[3].

However, as the coverage of scRNA-seq data is generally insufficient for consistently calling nucleotide variants, targeted amplification of loci containing somatic and mtDNA mutations is usually required[4,5]. Existing single-cell genotyping approaches include plate-[4] and droplet-based protocols[5–8], some of which entail adding locus-specific primers of predefined targets at the very initial steps of sample processing, namely, the stage of oil encapsulation. As all these approaches are based on Illumina short-read sequencing, they require large numbers of primers for covering all possible mutational sites across entire genes or genetic regions like the mitochondrial chromosome[7], creating cumbersome and inefficient experimental workflows. While Illumina currently provides the lowest cost per base at sequencing error rates below 1/1000, a clear limitation of short-read based sequencing is that it is suboptimal for long fragments, which creates considerable difficulties for detecting mutation sites that require amplicons exceeding a length of 500 nucleotides[9]. Additionally, short-read sequencing is unable to easily resolve structural transcriptomic variants such as gene fusions, transgenes or even isoforms that characterize malignant and immune cell subpopulations. Finally, single-cell analyses often reveal unanticipated genetic and transcriptomic variants whose detection would improve analytical resolution, thus creating the need to iteratively interrogate single-cell cDNA for such features even following library preparation.

Herein, we take advantage of the recent improvements in the read accuracy and throughput of long-read sequencing[10,11], which provides full-length coverage of single cell transcriptomes and hence reduces the complexity for the detection of genetic and transcriptomic variants. PacBio and Oxford Nanopore Technologies (ONT) are two established long-read sequencing platforms. PacBio was the first long-read sequencing technology to achieve low error rates comparable to Illumina[12]. However, with the recent introduction of the V14 chemistry, sequencing accuracy of ONT now exceeds 99%[13], while also having lower sequencing costs and providing a wider range of available flow cell sizes.

For these reasons, we provide a long-read based pipeline for the ONT platform using limited primer sets to amplify target genes from 5'-anchored 10x Genomics scRNA-seq whole-transcriptome cDNA libraries, thus enabling the flexible detection of a wide range of barcodes from single cell libraries, without spike-in of gene-specific primers during cDNA library preparation. We demonstrate that this approach can detect natural barcodes with sufficient accuracy for reliable lineage- and immune cell-tracing in numerous contexts, including the setting of ipilimumab-based immunotherapy for relapsed acute myeloid leukemia (AML) following allogeneic hematopoietic stem cell transplantation (HSCT)[14]. Altogether, we present a long-read sequencing-based framework for integrative genotyping of single cell profiles that substantially improves the resolution of leukemia and immune cell phenotypes.

## Results

### nanoranger: long-read sequencing-based genotyping of single cell RNA profiles

We developed a versatile workflow that enables the amplification, long-read sequencing, and processing of targets of interest using the ONT platform such that a wide range of natural barcodes, including somatic and mtDNA mutations, fusion genes and isoforms can be detected (Fig. 1a). The pipeline originates from single cell cDNA libraries that are whole-transcriptome amplified "intermediate libraries". To enrich for detection of various natural barcodes, we devised a 3-step PCR protocol for targeted amplification designed to capture the molecular feature of interest using a streamlined set of primers

(Fig. 1b; Supplementary Tables 1–7). In brief, this process entails first a clean-up PCR ("PCR 1"), in which shorter amplicons arising from template-switch oligo (TSO) artifacts are depleted from cDNA libraries by amplification with a generic 5' handle primer and a biotinylated generic 3' handle primer followed by streptavidin/biotin selection[15]. Loci of interest are then amplified with a generic 5' handle primer and a locus-specific 3' biotinylated primer using RNase H-dependent (rh) PCR[16] to enhance specificity of primer pairing with lowly expressed targets ("PCR 2"). Finally, following a second streptavidin/biotin selection, a third nested PCR ("PCR 3") generates material sufficient to proceed to ONT sequencing.

To integrate single cell gene expression profiles with the genotyping features of interest, we devised a processing pipeline called *nanoranger*. *Nanoranger* extracts cell barcode and transcript information from the ONT sequenced data and provides genome alignment for the calling of molecular features of interest. Because ONT generates naturally occurring multimer reads that contain multiple transcripts, we tuned *nanoranger* such that it can deconcatenate multimers, including transcripts with opposing orientations (i.e., 5' to 3' [molecule1] vs. 3'-to-5' [molecule 2], Fig. 1c). In doing so, the number of detected transcripts is increased, compared to previous approaches[17] that discard such reads due to reliance on error-prone identification of internal adapters, and which in turn can lead to incorrect assignment of barcodes to transcripts.

We performed a series of analyses to confirm the contributory role of the various components of this multi-step workflow. First, we verified that the strategy of depleting TSO priming artifacts[15] led to markedly improved coverage of transcripts and, therefore, improved genotyping, as shown for *TP53* and *RUNX1* transcripts (Fig. 1d). Second, we benchmarked the ability of *nanoranger* to deconcatenate multimer reads by comparison with *longbow*, an established deconcatenation tool for MAS-ISO-seq data which is generated from programmable adapter-based ligation of cDNA molecules[15]. To do so, we generated a library containing artificial 15-mers of mitochondrial transcripts using MAS-ISO-seq (Fig. 1e-top) and sequenced the library on ONT, followed by deconcatenation of the sequencing data with either *longbow* (tuned for the MAS-ISO-seq protocol) or *nanoranger* (Fig. 1e-bottom). The sequencing coverage distribution and number of extracted transcripts with *nanoranger* and *longbow* (10.7 vs. 8.5 million mapped reads) were similar, confirming *nanoranger's* ability to deconcatenate multimer reads. Small differences were attributable to the fact that *nanoranger* performs deconcatenation based on transcript alignment, whereas *longbow* is optimized for known adapter sequences. Third, to systematically assess the contribution of read deconcatenation to the performance of *nanoranger*, we analyzed a series of 13 ONT-sequenced amplicon libraries not deliberately concatenated using MAS-ISO-seq. A median of 22% ONT reads contained multiple extractable transcripts (range 3-42%) with up to 478 transcripts extracted from a single read. This yielded a median of 26% more transcripts than sequenced reads (range 3-49%) (Supplementary Fig. 1a–d).

Finally, we assessed *nanoranger's* performance for deconcatenating and quantifying MAS-ISO-seq data from healthy donor peripheral blood mononuclear cells sequenced with PacBio. When comparing the number of segments deconcatenated per read, *nanoranger* found consistently fewer segments (median 12; range 0–37) than the PacBio processing tool *skera* (median 15; range 0–16), yielding a total of 85,387,903 and 110,127,015 segments (Supplementary Fig. 1e, f). This is because *nanoranger* identifies only segments that align to the reference transcriptome (gencode v44) and therefore does not recognize non-human transcripts, genomic contamination, intronic or unannotated transcripts, such as repeat elements. Nevertheless, the mean number of detected molecules per gene (141 vs. 55) and per cell (2580 vs. 1650) were highly correlated and consistently higher with *nanoranger* compared to *skera* (r = 0.88 and 0.97) (Supplementary Fig. 1g, h), likely due to differences in the underlying reference

transcriptome and the annotation method. Similarly, *nanoranger* identified genes in more cells (122 vs. 50) and more genes per cell (1380 vs. 867) ($r = 0.87$ and 0.96) (Supplementary Fig. 1i, j). This resulted in identification of very similar cell types between the two analytical pipelines but better capture of immunologically relevant genes such as *HLA-E*, *IGHM* or *IL17RA* with *nanoranger* (Supplementary Fig. 2a, b). Altogether, these analyses demonstrate the generation of an analytical workflow optimized to maximally extract transcripts from ONT single cell data.

### Detection of natural genetic barcodes by ONT sequencing: efficiency and limitations

To assess the efficiency of extracting molecular features from long versus short-read sequencing data, we compared the highly polymorphic T cell receptor (TCR) sequences obtained from ONT and from Illumina sequencing. We re-sequenced a TCR cDNA library generated from melanoma-infiltrating T cells[18] (originally processed with the 10x Genomics (V)DJ kit) using both the Illumina and ONT sequencing platforms (Fig. 2a, Supplementary Fig. 3a, Supplementary Table 8). In the case of ONT, fragmentation of enriched TCR libraries was not needed, and hence provided a simpler workflow. The number of reads per CDR3 and the number of cells per CDR3 obtained from either sequencing platform were highly concordant (5653 [ONT] and 5767 [Illumina] cells with TCR detected) (Fig. 2b). Those cells without detected TCR sequences (755 [Illumina] and 787 [ONT]) were predominantly contaminant monocytes (Fig. 2c). The data also provided the means to estimate the sequencing performance of V14 ONT chemistry and Illumina. By comparing TCR reads against their respective consensus sequence, we observed per-base mismatch (0.54% [Illumina] vs. 0.83% [ONT]) and indel rates (0.08% [Illumina]

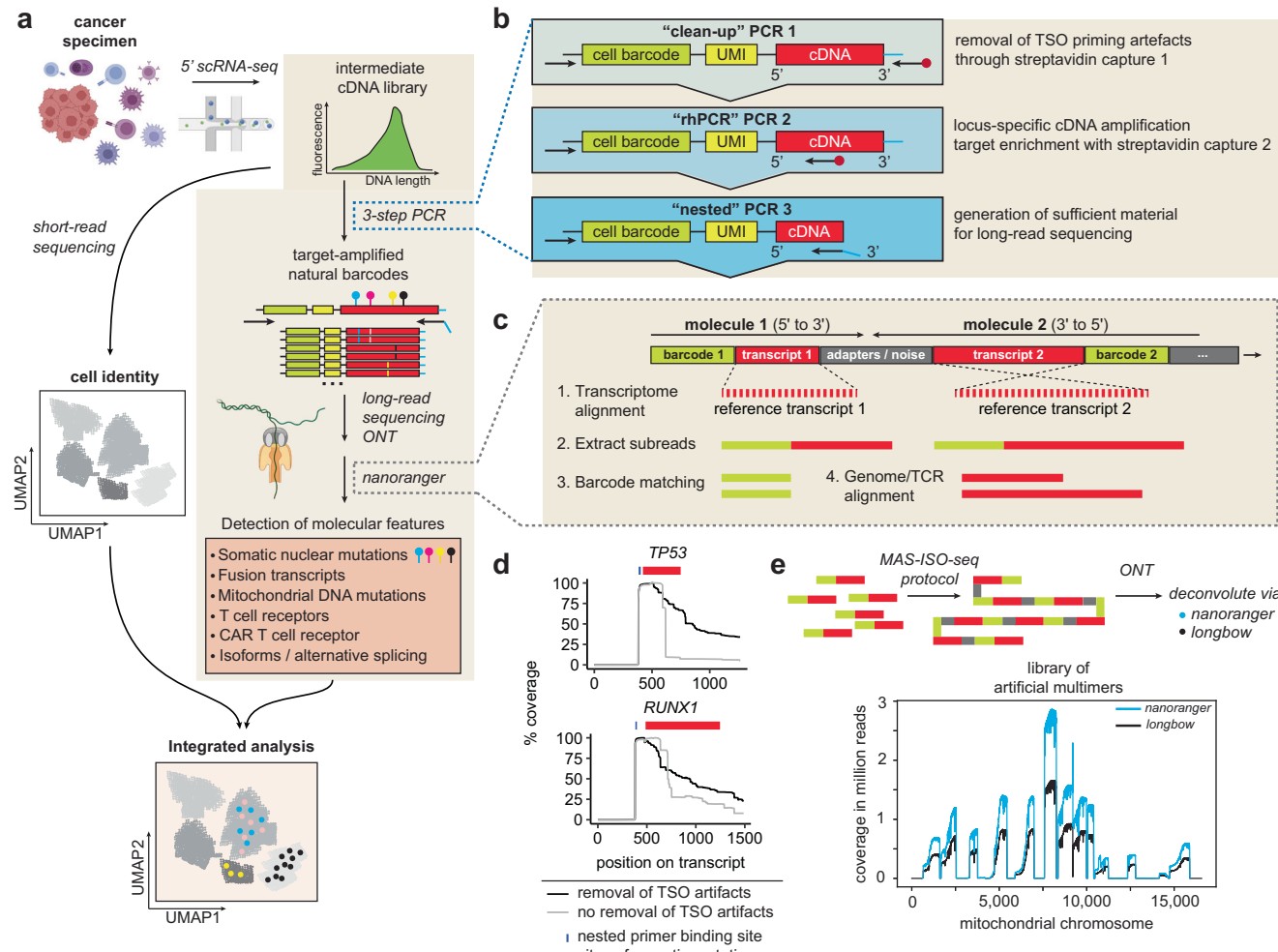

**Fig. 1 | *nanoranger* processes amplicons from 10x cDNA sequenced with the Oxford Nanopore platform. a** Intermediate single-cell cDNA libraries of cancer specimens are used for targeted amplification of transcripts carrying natural barcodes. By sequencing long amplicons on the Oxford Nanopore platform multiple genetic barcodes such as somatic nuclear mutations can be detected from the same amplification product. Applications for long-read sequencing include the read-out of T cell receptor (TCR) or CAR sequences, somatic nuclear and mitochondrial DNA mutations, fusion transcripts, and alternative splicing events (gene isoforms). The genotyping information from long-read sequencing is integrated with the gene expression data from short-read sequencing. **b** Amplicons are generated using a 3-step PCR. In the first PCR template-switch oligo (TSO) artifacts are removed with generic amplification of cDNA using a biotinylated 3′ primer and streptavidin purification. In the second PCR, gene-specific biotinylated 3′ primers are used to amplify loci of interest. After a second streptavidin purification, target genes are amplified with nested gene-specific 3′ primers to provide sufficient material for sequencing. **c** Overview of the *nanoranger* workflow. Multimer reads are decon-catenated by identifying transcripts with alignment against a reference transcriptome (1). After extraction of subreads (2), cell barcodes are identified (3) and TCR information is processed or transcripts are genome-aligned (4) for downstream genotyping. **d** Examples of gene coverage with (black) and without (gray) removal of TSO artifacts. The blue line indicates the primer binding site and the red ribbon shows locations of mutations used for lineage tracking in AML detected with each primer set. **e** Benchmarking of multimer demultiplexing using *nanoranger* and *longbow* on artificially generated multimers using the ISO-MAS-seq protocol. Of note, *nanoranger* deconcatenates multimer reads agnostic of adapters between transcripts, while *longbow* is optimized for known adapter sequences.

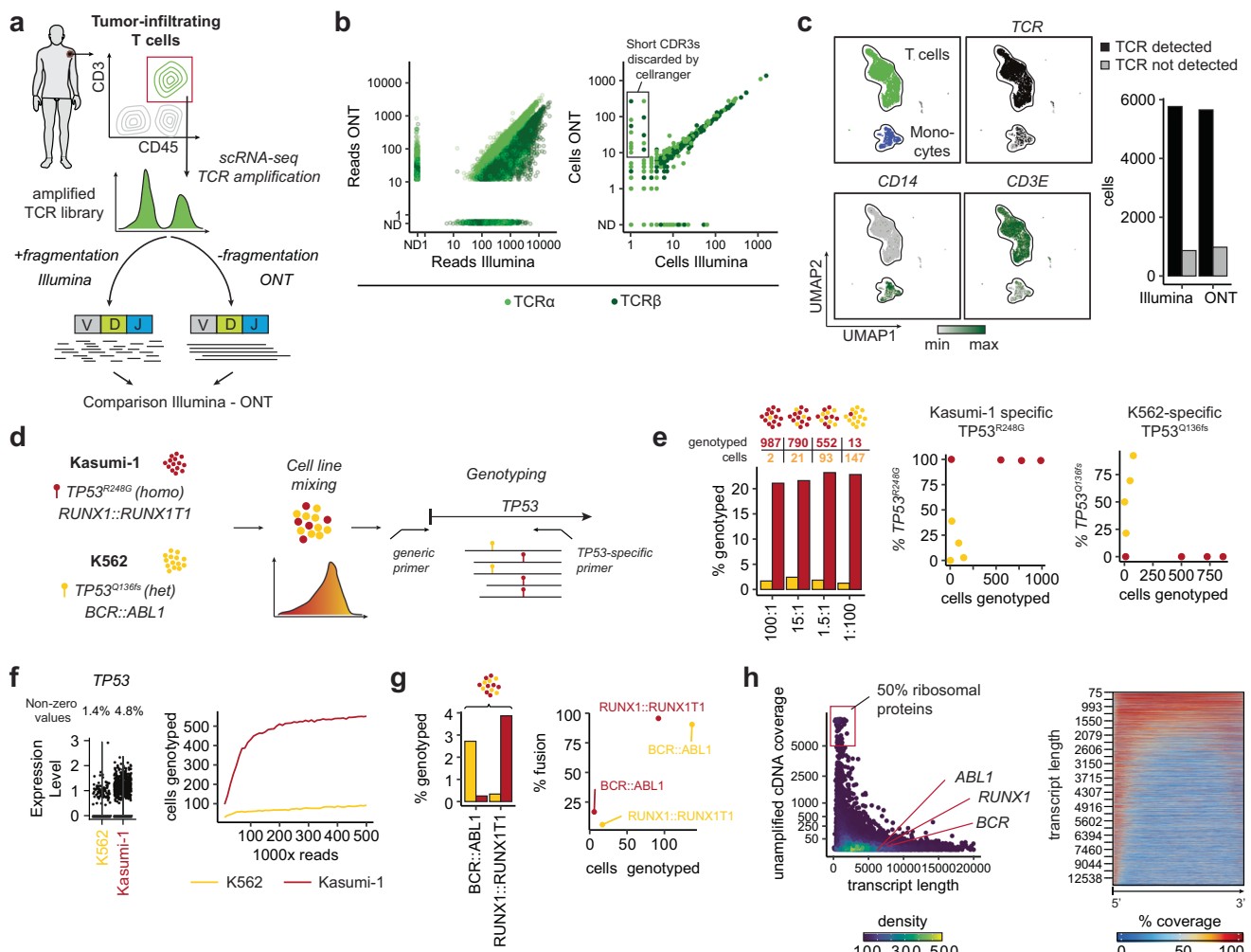

**Fig. 2 | *nanoranger* provides accurate cell barcode and genotyping information. a** Tumor-infiltrating T cells from one melanoma case (patient C described in Oliveira et al., *Nature 2021*) were isolated and used for single cell sequencing. After generation of enriched T cell receptor (TCR) libraries, they were either fragmented and sequenced with Illumina (left) or directly sequenced without fragmentation with Oxford Nanopore (ONT) (right). **b** Comparison of reads obtained with Illumina and ONT (*nanoranger*) per cell barcode shown for TCRα (light) and TCRβ (dark) sequences (left) or number of cells obtained with Illumina and ONT with a particular CDR3α (light) or CDR3β (dark) (right). The rectangle indicates unproductive CDR3s filtered by cellranger. **c** UMAP representation of cell types (top left), cells with detectable TCR (top right), expression of *CD14* and *CD3E* (bottom). The bar plot demonstrates the number of TCR sequences obtained by Illumina and ONT sequencing. **d** Overview mixing experiment with Kasumi-1 (acute myeloid

leukemia, AML) and K562 (chronic myeloid leukemia, CML) cells. Kasumi-1 contain a homozygous *TP53^R248G* mutation and the *RUNX1::RUNX1T1* fusion gene. K562 express only one *TP53* allele with a truncating frameshift mutation (*TP53^Q136fs*) and *BCR::ABL1*. Both *TP53* mutations are detected with the same primer. **e** Absolute number and percentage of genotyped Kasumi-1 (red) and K562 cells (yellow) for the *TP53^R248G* mutation (left) and percentage of cells carrying either *TP53^R248G* or *TP53^Q136fs* (right). **f** Expression of *TP53* and percentage of cells with detectable *TP53* transcripts (left). Number of cells genotyped as function of the number of *TP53* reads shown for Kasumi-1 (red) and K562 (yellow) in a downsampling experiment (right). **g** Percentage of genotyped cells for *BCR::ABL1* and *RUNX1::RUNX1T1* (left). Percentage of cells with *BCR::ABL1* and *RUNX1::RUNX1T1* shown for Kasumi-1 (red) and K562 (yellow) (right). **h** Coverage across all detectable genes from a 10x Genomics cDNA library after removal of TSO artifacts as function of transcript length.

and 0.25% [ONT]) that were slightly higher with ONT ($p < 0.001$) (Supplementary Fig. 3b–d). Consistent with known characteristics of ONT[10,19,20], the indel rate increased in reads with homopolymers such as guanine-repeats (Supplementary Fig. 3e, f). Based on consensus sequences that overcome these sequencing errors, *nanoranger* nevertheless reliably extracted highly polymorphic TCR sequences and the highly correlated results of single cell TCR clonotypes with short- and long-read sequencing ($r = 0.8$ [CDR3α] and 0.9 [CDR3β]) demonstrate the sufficient accuracy of the ONT-based workflow for detecting such natural genetic barcodes.

To assess the specificity and sensitivity of our protocol to identify recurrent somatic mutations from tumor cells, we performed mixing studies of two leukemia cell lines with distinct point mutations and gene fusions. We mixed Kasumi-1 (an AML line harboring *AML1::ETO* and homozygous *TP53^R248G*) with K562 (a chronic myeloid leukemia

[CML] line harboring *BCR::ABL1* and monoallelic *TP53^Q136fs*) cells at four defined ratios (100:1, 15:1, 1.5:1, 1:100) (Fig. 2d); at each ratio, we amplified and sequenced the homozygous *TP53^R248G* mutation found in Kasumi-1 cells[21]. Compared to a 0% genotyping rate from native scRNA-seq gene expression data, the number of Kasumi-1 cells that could be genotyped with *nanoranger* ranged from 13 to 987, with a consistent genotyping rate of ~22% across the four samples analyzed (Fig. 2e-left). Among the Kasumi-1 cells with amplified *TP53* segments, the *TP53^R248G* mutation was detected in 98.8–100% of cells, indicating the high specificity of our assay (Supplementary Fig. 4a). In contrast, K562 cells, which have lost their second *TP53* allele[22], and hence have much lower levels of *TP53* expression, generated genotyping rates of only 1.2–2.8% compared to the null detection of this mutation in Kasumi-1 cells (Fig. 2e-right, Fig. 2f-left, Supplementary Fig. 4b, c). The downsampling analysis of the *TP53* reads from both cell lines (Fig. 2f-right) showed

that mutation detection was well above saturation. Thus, the targeted amplification of the *TP53* gene captured most of the *TP53* molecules present in the original cDNA library, and the limited number of genotyped cells was not due to inadequate sequencing depth but rather from differences in the native expression level of the targeted genes within these cell populations[23].

Likewise, despite the high sensitivity and specificity of our workflow, our detection rate of the cell line- specific fusion genes was low: only 124 (2.7%) K562 cells with *BCR::ABL1* and 88 (3.9%) Kasumi-1 cells with the *RUNX1::RUNX1T1* fusion transcript (Fig. 2g). To investigate this low detection rate, we analyzed the absolute coverage across all detectable transcripts in a 5′ 10x Genomics cDNA library sequenced with ONT after removal of TSO artifacts (clean-up PCR). While coverage was highest for short transcripts (<1000 bp) of ribosomal proteins and other ubiquitously expressed genes like *HLA-B/C* or mitochondrial genes, mean coverage dropped quickly for longer transcripts (Fig. 2h-left). Across all transcripts, irrespective of their absolute length, the coverage was highest within the first 4000 bases at the 5′ end and dropped quickly towards the 3′ end (Fig. 2h-right; Supplementary Fig. 4d), such that transcripts with absolute length greater than 4000 bases show distinctly reduced coverage. Consistent with this observation, most amplified reads mapping to *BCR::ABL1* were not full-length but ranged in length from 100 to 500 bp (Supplementary Fig. 4e). Thus, a likely reason for the low detection rate of fusion transcripts in these experiments was the distance of the fusion breakpoints from the 5′ end. Together, ONT-based long-read sequencing of targets amplified from intermediate 10x Genomics cDNA libraries appears to be best suited for targets on highly expressed, short transcripts within the first 4000 bp of the 5′ end. Moreover, for lesions residing close to the 3′ end, we anticipate that utilization of 3′ chemistry would be preferable, as this would have better coverage within the last 4000 bp of each transcript (Supplementary Fig. 5a).

### nanoranger and GoT capture distinct cell barcodes

To benchmark performance of *nanoranger* to Illumina-based mutation detection, we processed a bone marrow sample of an AML case with three somatic mutations (*DNMT3A*[R882H], *RUNX1*[I177S], *SF3B1*[K700E]) at relapse after allogeneic HSCT with *nanoranger* and with the 5′ genotyping of transcriptomes (GoT) protocol[5] (Fig. 3a). In addition to the Illumina sequencing described in the published protocol, we sequenced the unfragmented GoT library on ONT, processing the raw data with the *nanoranger* analytical pipeline. Overall, *nanoranger* and GoT sequenced with Illumina achieved similar genotyping rates (4.8–19.2% for nanoranger; 5.5–19.1% for GoT) of scRNA-seq profiles for two of the three mutations (*DNMT3A*[R882H], *SF3B1*[K700E]) that we targeted (Fig. 3b, c). For *RUNX1*[I177S], located in close proximity to the 5′ end, GoT reverse transcriptase (RT) primers improved capture from 15.1% to 39.2% (Supplementary Fig. 6a–c). For all three targets, we observed that the performance of GoT increased by another 1-7% when sequenced with ONT and processed with the *nanoranger* pipeline. Across all three experimental conditions, 99% cells with an identified *SF3B1*[K700E] mutation were recipient-derived (Methods), demonstrating the specificity of these approaches (Fig. 3d). The largest difference between data acquired with *nanoranger*, GoT sequenced on Illumina, and GoT sequenced on ONT was the cell barcode representation (Fig. 3e). We speculated that this could be in part due to a lower capture efficiency of longer library fragments with Illumina sequencing. We therefore analyzed the minimal fragment length in reads from the GoT library that associated with cell barcodes identified with Illumina and ONT sequencing versus those that were only identified using ONT sequencing. This revealed that Illumina sequencing did not capture fragments that were longer than -1.5kB, demonstrating the advantage of long-read sequencing for genotyping of loci that are not immediately adjacent to the 3′ or 5′ of a transcript (Fig. 3f).

In sum, *nanoranger* has comparable performance to GoT, but genotypes different cell barcodes due to differences in sequencing capture rates of longer library fragments. The GoT ONT results with *nanoranger* processing indicate that including gene-specific RT primers during the cell encapsulation step can improve the genotyping rate of targets close to the 5′ end but requires the prescience to select targets prior to initiation of a single cell project (Supplementary Fig. 6d). As illustrated by the numerous examples presented herein, the full *nanoranger* workflow enables re-analysis of archived cDNA libraries so that targets can be flexibly added to address new hypotheses that are generated after the initial single cell analysis.

### Tracking mutated and non-mutated hematopoietic cell lineages in AML

AML-associated somatic mutations have been previously described across the myeloid differentiation trajectory, leading to the description of 6 AML gene expression clusters (EC: HSC, Progenitor, GMP, Promono, Mono, and cDC)[4]. We confirmed the detection of these AML ECs through the re-analysis of a recently reported scRNA-seq analysis of serial bone marrow samples obtained from study participants enrolled in the phase I ETCTN/CTEP 10026 study that tested combined decitabine and ipilimumab treatment in two patient cohorts: transplant-naïve AML/MDS or post-allogeneic HSCT (Fig. 4a)[24]. Distinguishing normal and malignant hematopoiesis using expressed donor- and recipient-specific single nucleotide polymorphisms (SNPs) (*souporcell*)[25] revealed two additional AML-derived ECs, megakaryopoiesis and erythropoiesis, to be almost entirely recipient-derived in 6 of 8 analyzed cases from the post-HSCT cohort at time of relapse prior to initiation of decitabine and ipilimumab (non-fractionated bone marrow chimerism 2–85%) (Fig. 4b-left; Table 1). To extend this analysis, we used *nanoranger* to genotype individual AML/MDS cells originating from 9 patients enrolled in either of the two cohorts, targeting 11 recurrently mutated AML/MDS-associated genes (*ASXL1, DNMT3A, NRAS, IDH2, RUNX1, SF3B1, SRSF2, STAG2, TET2, TP53, U2AF1*) (Supplementary Table 4). From 18,097 genotyped profiles (median of 610 genotyped cells per sample, range 180–5507), most AML-associated mutations were detectable across myeloid progenitor, monocytic, and dendritic cell populations (Fig. 4b-right, Supplementary Fig. 7a, Supplementary Tables 9, 10). Of note, we also identified AML-associated mutations in the erythroid and megakaryocytic progenitor populations in all samples, indicating these two cell compartments directly differentiate from leukemic clones. CNV changes associated with several AML cases likewise segregated in not only myeloid but also erythroid and megakaryocytic progenitor cell populations (Supplementary Fig. 7b–d). A further elucidation of leukemic subclonal structure was provided by the example of AML1019, where we found two separate AML clones that were not only defined by presence or absence of *RUNX1*[R320*], but also by *amp(8p)* and *amp(21p)* (Supplementary Fig. 7d).

Throughout our analysis of the AML cohort, we repeatedly noticed samples with apparently homozygous mutated and wild-type cells at roughly equal proportions, indicating sampling only one transcript per cell. For example, analysis of AML1022 (Supplementary Fig. 8a-left) revealed that only one UMI was captured in most cells (77–93%), despite mean coverage of locus-specific reads per cell ranging from 36 to 248 (Supplementary Fig. 8a-right, 8b). Therefore, most observations of apparent homozygosity are probably due to allelic drop-out. Concordantly, we observed aggregated VAFs of 50% while pseudobulk donor chimerism was 0% (Supplementary Fig. 8c, d). Thus, we caution that while detection of somatic mutations was specific for leukemic clones, their absence is not sufficient to identify wild-type cells.

To evaluate the longitudinal dynamics of AML clones, we obtained genotyping information at screening and at time of response for 5 AML/MDS cases who achieved complete remissions. In all cases,

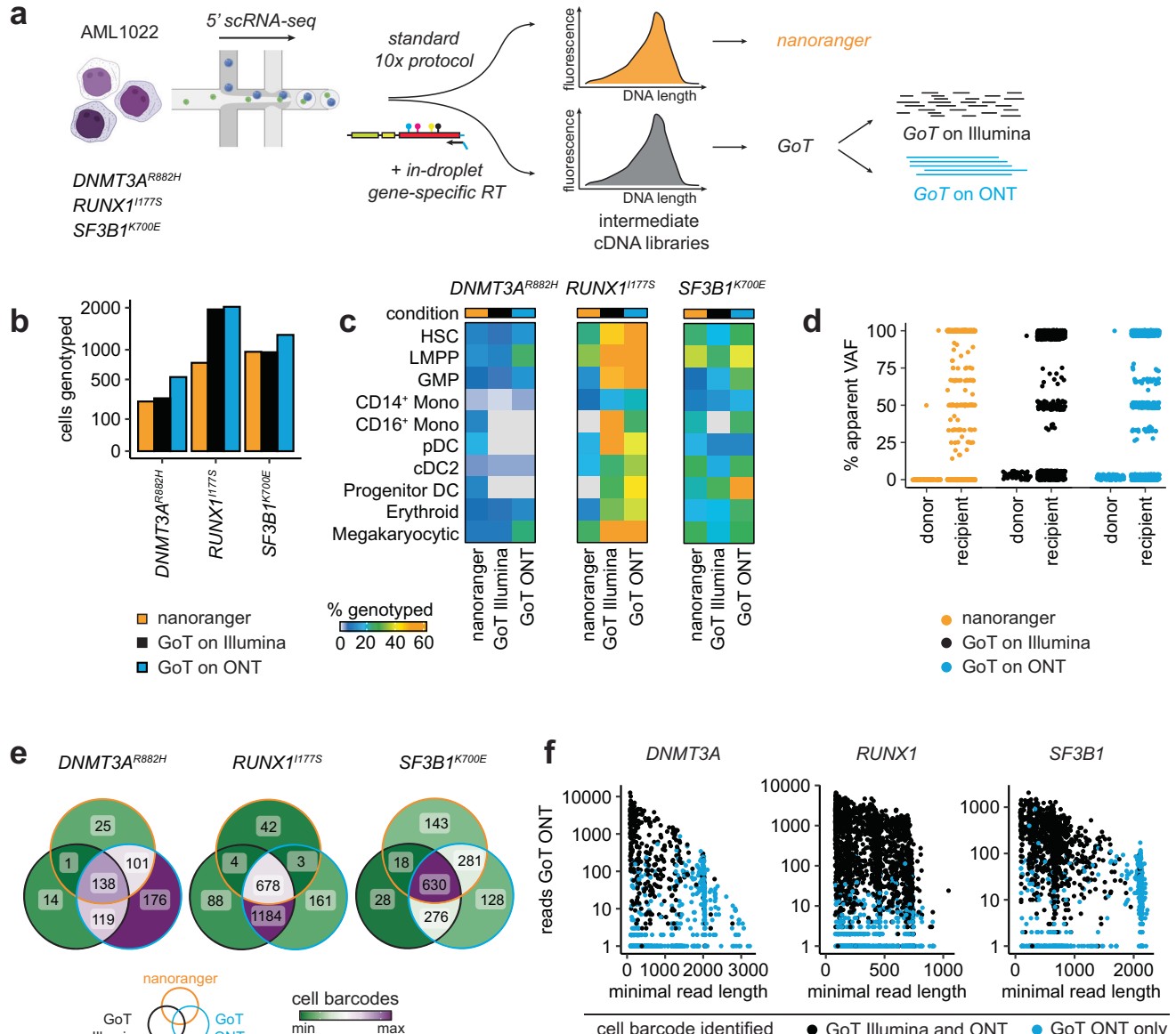

**Fig. 3 | Comparison of *nanoranger* with GoT. a** Experimental workflow of comparison between *nanoranger* and genotyping of transcriptomes (GoT). A pre-treatment bone marrow sample of AML1022 at relapse after allogeneic hematopoietic stem cell transplantation (HSCT) was used for single cell cDNA library preparation according to the standard 10x Genomics 5′ gene expression protocol and following the modified 5′ GoT protocol with in-droplet inclusion of gene-specific reverse transcriptase primers. Both cDNAs were taken forward for sequencing with the standard *nanoranger* protocol (orange), GoT using Illumina sequencing (black) and GoT using Oxford Nanopore sequencing (blue). **b**, **c** Number of cells genotyped with each experimental condition (b) and percentage of genotyped cells across hematopoietic differentiation states (c).

**d** Comparison of apparent single cell variant allele frequencies (VAFs) for *SF3B1^K700E^* in donor- versus recipient-derived cells to demonstrate specificity of genotyping with each experimental condition. **e** Comparison of cell barcodes identified with each condition. The venn diagrams demonstrate the number of cell barcodes that are uniquely identified or shared across experimental conditions. To enable direct comparison of captured cell barcodes, the cDNA for the GoT condition was used as input for *nanoranger*. **f** Minimal read length versus number of reads for cell barcodes identified with GoT on Illumina and ONT (black) versus those identified only with GoT on ONT (blue), demonstrating the preferential sequencing of shorter fragments with Illumina sequencing.

---

somatic mutations remained detectable in HSC, LMPP, GMP, megakaryocytic and erythroid cells over time (Fig. 4c, Supplementary Fig. 9a), unambiguously revealing that the cytoreductive effect of therapy was not accompanied by leukemic clone eradication (in fact, all patients subsequently relapsed)[14], which we previously could only infer on the basis of chimerism analysis[24]. In the case of AML8007, we detected evidence of a differential therapeutic response among subclones. At screening, three somatic mutations (*DNMT3A^V296M^*, *TP53^C176S^*, *TP53^R282W^*) and three chromosomal aberrations (*amp(1p)*, *del(3p)* and *del(5q)*) were detected (Fig. 4d, e). At remission, the somatic mutations and *del(5q)* remained detectable, but *amp(1q)* and *del(3p)* became

undetectable in all cell compartments except for LMPP (Supplementary Fig. 9b, c). Together, this supports *del(5q)* as the founder lesion, followed by acquisition of somatic mutations in *DNMT3A* and *TP53*, and finally a subclonal event with acquired *amp(1q)* and *del(3p)* associated with differential sensitivity to decitabine and ipilimumab treatment.

Comparison of the expression features of AML clones revealed that none of the signatures of the previously reported ECs showed high expression scores in erythroid and megakaryocytic cells (Supplementary Fig. 10a–c). We thus devised 5-gene expression signatures for these two cell types consisting of *GATA1, CA1, HBA1, HBB, ALAD*

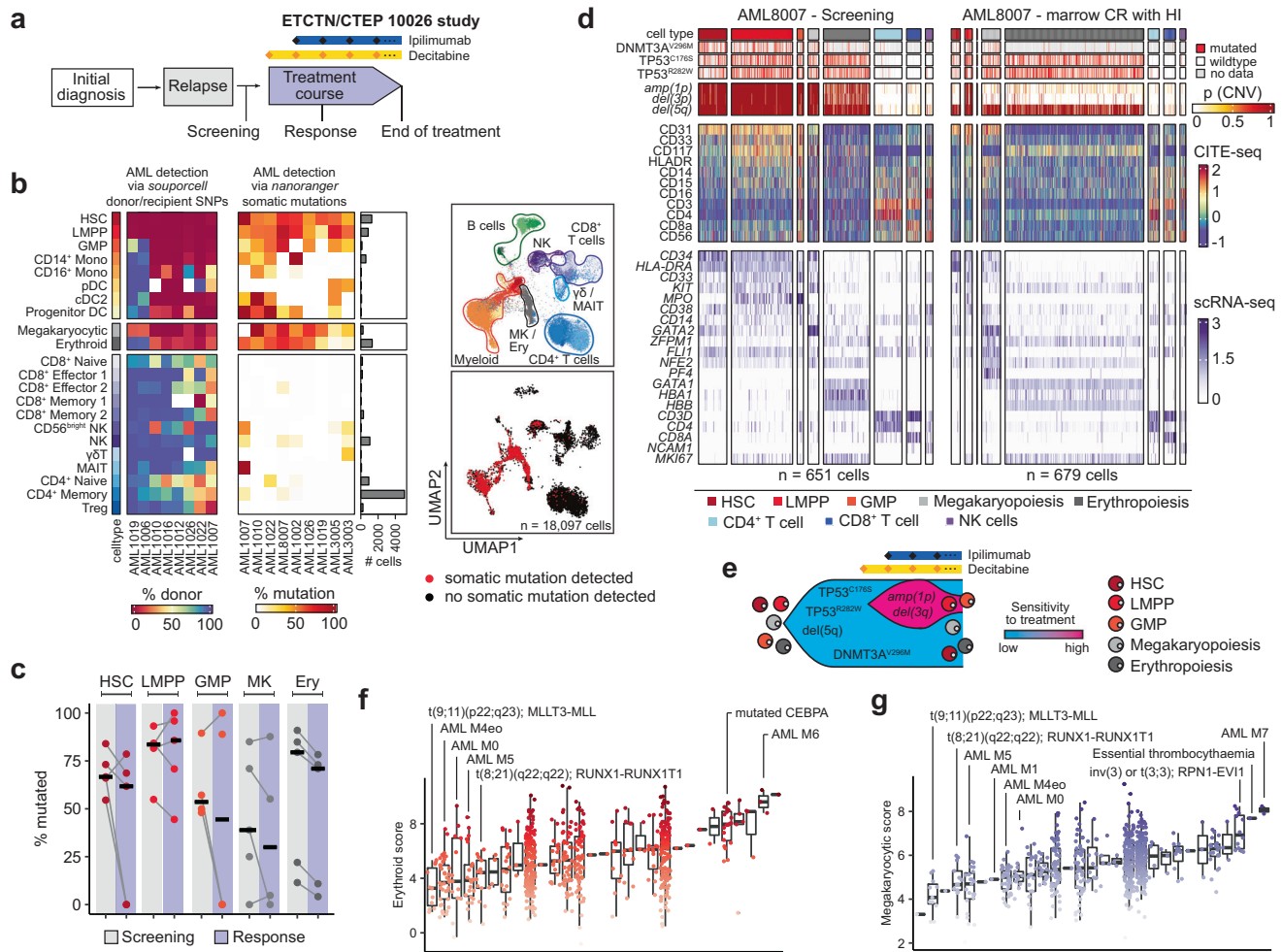

**Fig. 4 | AML-associated somatic mutations are detectable in myeloid, erythroid and megakaryocytic progenitor populations. a** AML bone marrow aspirates for single cell RNA sequencing (scRNA-seq) were obtained from participants in the ETCTN/CTEP 10026 study, which tested combined decitabine and ipilimumab in advanced MDS/AML or relapsed AML post-HSCT patients. Samples were analyzed at study entry (screening) and time of response. **b** Percentage of donor-derived cells across bone marrow associated immune cell types across 8 AML cases re-analyzed from Penter et al., *Blood 2023*[24] using *souporcell* (left). Percentage of genotyped cells with AML-associated somatic mutation for bone marrow-associated hematopoietic cell types across 9 AML cases (right). Cell types from scRNA-seq profiles were identified using short-read sequencing and reference-based annotations (UMAP top). Recurrently mutated genes were amplified and sequenced using *nanoranger* to identify leukemic cell compartments and immune cells with recurrent AML-associated mutations (UMAP bottom). **c** Detection of recurrent somatic mutations at screening and at time of response in 5 study participants of ETCTN/CTEP 10026 across HSC ($n = 847$), LMPP ($n = 1,198$), GMP ($n = 154$), megakaryocytic (MK) ($n = 815$) and erythroid (Ery) ($n = 3,223$) cells. **d** Heatmap of somatic mutations ($DNMT3A^{V296M}$, $TP53^{C176S}$, $TP53^{R282W}$), chromosomal aberrations ($amp(1p), del(3p), del(5q)$) visualized as joint posterior probability calculated with *numbat*, protein expression and gene expression in AML8007 at screening and at time of response (marrow CR with HI—bone marrow complete remission with hematologic improvement). **e** Illustration of likely subclonal structure of AML8007 derived from (**d**) as fish plot with high (magenta) and low (blue) sensitivity to decitabine and ipilimumab. Expression of erythroid (**f**) and megakaryocytic (**g**) signature in 646 Beat AML bulk RNA-seq profiles of AML/MDS subtypes. Boxes represent the interquartile range (IQR) from the 25th to 75th percentile and the median value indicated by the inner horizontal line. Whiskers extend to the extreme values but no more than 1.5xIQR above/below the hinge.

(erythroid) and *GATA2, ZFPM1, FLI1, NFE2, PF4* (megakaryocytic). When applied along with the previously defined EC gene signatures to a large public dataset consisting of 646 AML bulk RNA-seq profiles (Beat AML)[26], these two signatures scored highest in essential thrombocythemia, and acute erythroid and megakaryoblastic leukemia. We also saw a high erythroid score in AML with mutated *CEBPA*, consistent with observations of expanded mutated erythroid cells in AML harboring double mutant *CEBPA*[27]. The megakaryocytic score was high in AML with *inv(3)* or cases carrying a *RPN1-EVI1* fusion, also consistent with known megakaryocyte-lineage skewing in this subtype (Fig. 4f, g, Supplementary Fig. 10d)[28]. To determine if recurrent somatic mutations in erythroid and megakaryocytic leukemic populations were detectable in the data used to define the original AML ECs, we performed a re-analysis and indeed identified recurrent somatic mutations in these compartments, but the overall number of genotyped

erythroid or megakaryocytic progenitors was very low (<10) for most samples (Supplementary Fig. 10e), precluding statistical significance.

Altogether, our integrated analysis of recurrent somatic mutations, donor chimerism and copy number changes at single cell resolution revealed not only clear definition of individual AML clones present in myeloid cellular compartments but also their differentiation into erythroid and megakaryocytic lineage in the setting of relapsed/refractory secondary AML, consistent with two recent single-cell sequencing studies in AML/MDS[8,29]. These discovery findings provide leukemia-discriminating signatures that can yield more accurate analysis of bulk RNA sequencing profiles of AML.

## Mitochondrial DNA mutations for tracking AML
Mitochondrial DNA (mtDNA) mutations have recently been recognized as natural barcodes that can mark clonal cell populations and hence

**Table 1 | Clinical information study participants ETCTN/CTEP 10026**

| Study ID | Diagnosis | Mutations[a] | CNV changes from clinical karyotyping | Transplant status | % Bone marrow non-fractionated chimerism[b] | Genotyping performed |
|---|---|---|---|---|---|---|
| 1002 | sAML from essential thrombocythemia | $IDH1^{R132C}$ $IDH2^{R140Q}$ $JAK2^{V617F}$ $TET2^{I1873T}$ $TP53^{H179R}$ $TP53^{P278S}$ $U2AF1^{S34Y}$ | del(7q) | naïve | - | Yes |
| 1006 | sAML arising from MDS | $U2AF1^{S34Y}$ | del(20q) | post | 78 | No |
| 1007 | MDS EB-2 | $ASXL1^{D988fs*}$ $IDH2^{R140Q}$ $STAG2^{K1439*}$ | none | post | 31 | Yes |
| 1010 | AML | $NRAS^{G12D}$ | none | post | 2 | Yes |
| 1012 | AML | $TP53^{V173M}$ | inv(3) del(5q) add(21) | post | 11 | No |
| 1016 | tAML | - | del(5q) del(17p) | post | 84 | No |
| 1019 | sAML from MDS | $ASXL1^{Q748*}$ $RUNX1^{R320*}$ | amp(8) amp(21) | post | 85 | Yes |
| 1022 | sAML from MDS | $DNMT3A^{R882H}$ $RUNX1^{I177S}$ $SF3B1^{K700E}$ | del(13) del(19) del(20q) | post | 9 | Yes |
| 1026 | sAML from MDS/MPN overlap syndrome | $ASXL1^{G646Wfs*}$ $NRAS^{G13R}$ $SF3B1^{R775Q}$ | none | post | 6 | Yes |
| 3003 | MDS EB-2 | $ASXL1^{Y591*}$ $STAG2^{Q275*}$ $TET2^{N1504Kfs*}$ $TET2^{N488Mfs*}$ | none | naïve | - | Yes |
| 3005 | sAML from MDS | $DNMT3A^{N403Tfs*}$ $TP53^{R273H}$ $U2AF1^{S34Y}$ | del(5q) | naïve | - | Yes |
| 8007 | MDS EB-2 | $DNMT3A^{V296M}$ $TP53^{C176S}$ $TP53^{R282W}$ | amp(1q) del(3) del(5q) amp(6p) del(9) del(10) | naïve | - | Yes |

*AML* acute myeloid leukemia, *sAML* secondary AML, *tAML* therapy-related AML, *MDS* myelodysplastic syndrome, *MDS EB-2* MDS with 11–20% blasts.
[a]Mutational profiles were identified prior to this work using amplicon-based targeted sequencing of recurrently mutated genes in myeloid hematologic malignancies[52].
[b]Bone marrow bulk chimerism was determined immediately prior to treatment on ETCTN/CTEP 10026.

enable tracking of leukemia cell subpopulations[1,30], or even donor and recipient populations at single-cell resolution following HSCT (Fig. 5a)[31]. However, it remains unclear whether mtDNA and somatic mutations provide similar definitions of leukemic clones. As the native coverage of mitochondrial transcripts in cDNA varies by more than 10-fold and is largely insufficient for calling of variants, we developed a 20-primer panel (Supplementary Table 5) for their targeted amplification and subsequent long-read sequencing (Supplementary Fig. 11a–c). Abundance of mitochondrial transcripts appeared to be highest in metabolically active, proliferative AML blasts, which led to preferential amplification of their mitochondrial transcripts. Our data thus indicate that such cells are well-suited for detection of mtDNA mutations (Supplementary Fig. 11d, e).

To determine how well mtDNA mutations called from scRNA-seq libraries would correlate to mutation calls with mtscATAC-seq, an established ATAC-seq based single-cell protocol for the detection of mtDNA mutations[32], we processed the same bone marrow sample of AML1026 in parallel with both technologies and compared their pseudobulk VAFs. For mtDNA mutations with an overall VAF > 0.5%, both technologies were highly correlated ($r = 0.8$), indicating that such mutations likely represent reliable clonal markers (Supplementary Fig. 11f). We therefore integrated the detection of mtDNA and somatic

mutations for AML1026, who relapsed following HSCT, for the identification of donor or recipient-derived cells and longitudinal leukemic clone tracking. Eight mtDNA mutations (14766C > T, 14905G > A, 15452C > A, 15607A > G [donor] and 4011C > T, 8433T > C, 9722T > C, 15833C > T [recipient]) were used to distinguish donor and recipient-derived cell populations and were consistent with the annotation derived from analysis of expressed SNPs: 1855 recipient- and 532 donor-derived cells were annotated concordantly, with disagreement in only 14 cells (Fig. 4b, Supplementary Fig. 11g). Aside from individual-specific variants, mtDNA mutations were enriched in myeloid progenitor cells. For example, abundance of 10685G > A remained similar among myeloid progenitor cells, despite cytoreduction following treatment with decitabine and ipilimumab (Fig. 5c), consistent with observations made using somatic mutations. Further, we detected three somatic mutations ($NRAS^{G13R}$, $SF3B1^{R775Q}$, and $SRSF2^{P95H}$) that co-occurred with mitochondrial mtDNA mutations such as 10685G > A but were absent in T and NK cells (Fig. 5d, e).

In another notable example (de-novo AML1) (Table 2), we discovered two leukemic clones defined by distinct mutually exclusive mutations in *NPM1*. Clone 1 ($NPM1^{W287fs}$) also harbored FLT3-ITD and *loh(13)*, which were absent in clone 2 ($NPM1^{W288fs}$). By evaluating mtDNA mutations, we observed that both clones could be identified by a total

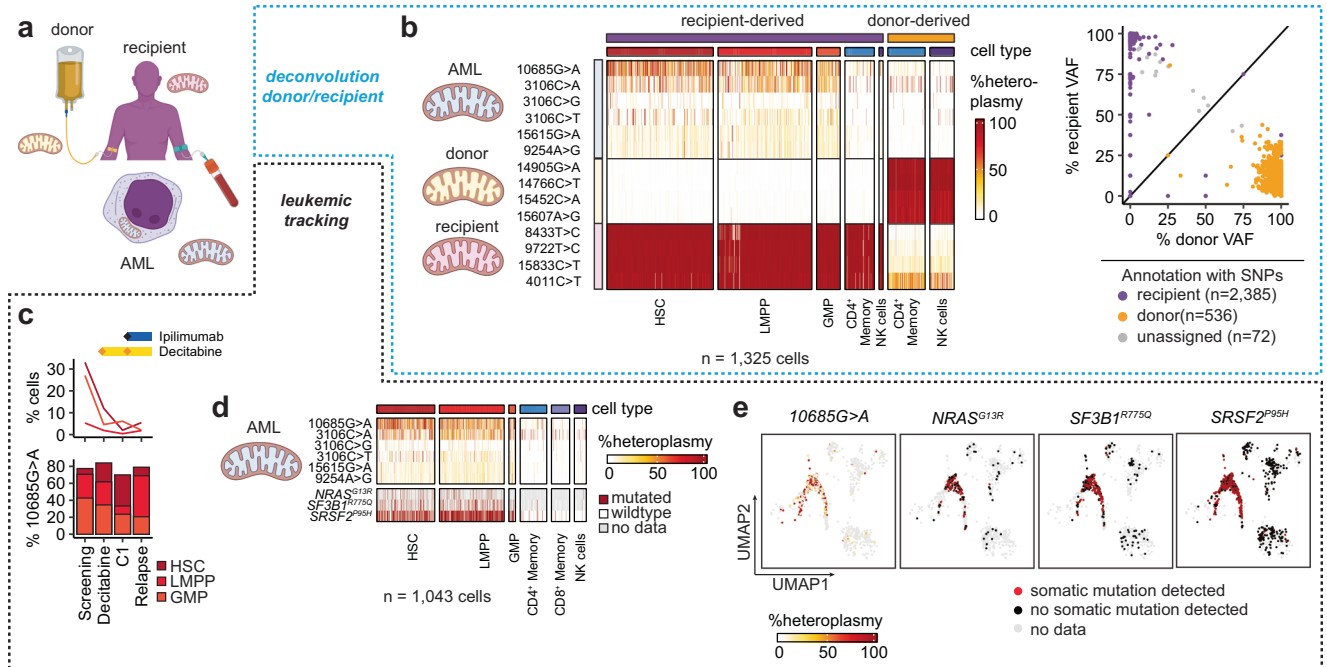

**Fig. 5 | Identification and tracking of leukemic cell populations using mitochondrial DNA mutations. a** Mitochondrial DNA mutations detected in scRNA-seq libraries with long-read sequencing can be used for identification of donor- and recipient-derived cells (top) and leukemic tracking (bottom). **b** Detection of donor- (yellow) and recipient-specific (purple) mtDNA mutations as well as mtDNA mutations enriched in recipient-derived progenitor populations (AML) (heatmap left). The mean variant allele frequency (heteroplasmy) of donor- and recipient-defining mtDNA mutations distinguish donor- and recipient-derived cells. As a validation, cells were also annotated as donor and recipient by *souporcell* which uses expressed single nucleotide polymorphisms (SNPs) for identification of individuals (right). **c** The 10685G > A mutation remains detectable in myeloid progenitor cells throughout treatment with decitabine and ipilimumab despite reduction in their frequency, suggesting this mutation to be a potential disease marker for this AML case. **d, e** Mitochondrial DNA mutations enriched in recipient-derived progenitor cells (HSC, LMPP, GMP) co-occur with somatic mutations (*NRAS*[G13R], *SF3B1*[R775Q] and *SRSF2*[P95H]), demonstrating their specificity for the AML.

## Table 2 | Patient and molecular features of de-novo AML cases

| Case | Age (Years)/sex | Clinical karyotype | Somatic mutations according to RHP[52] | FLT3-ITD sequence according to RHP[52] |
|---|---|---|---|---|
| de-novo AML 1 | 70/M | 47, XY, +4[3]/ 46, XY[17] | NPM1[W288fs] FLT3-ITD TET2[T606fs] | TCTAAATTTTCTCTTGGAAACTCCCATTTGAGATCATATTCATATTCTCTGAAATCAACGTCAAAC |
| de-novo AML 2 | 57/M | 46, XY[20] | DNMT3A[R882H] FLT3-ITD NPM1[W288Cfs*12] PTPN11[S502A] | TTCATATTCTCTGAAATCAACGTAG |
| de-novo AML 3 | 59/F | 47, XX, +8[8]/ 46, XX[12] | BCOR[R1164*] FLT3-ITD RUNX1[P294fs*] | ATTCTCTGAAATCTACGTAAG |

*F* female, *M* male, *RHP* rapid heme panel (amplicon-based clinical genetics)[52].

of 6 mutually exclusive mtDNA mutations (Fig. 6a–c). Both clones were further distinguished by their phenotypes: clone 1 differentiated along the entire myeloid, megakaryocytic, and erythroid trajectory, while clone 2 had a more confined progenitor-like phenotype. These findings were confirmed by analyzing the distribution of the mtDNA mutations. Within GMP-like cells, further gene expression differences between both clones of de-novo AML1 could be identified such as differential expression of myeloid markers like *LYZ* or *CST3* (Fig. 6d, e). This case demonstrates that, like secondary AML and MDS, de-novo AML can also differentiate from HSC-like to monocytic or even megakaryocytic and erythroid populations, which we also observed in two additional cases of de-novo AML (Supplementary Fig. 12a, b). Together, our data reveal that mtDNA mutations can distinguish donor and recipient-derived cells with high accuracy, may serve as alternative disease markers in AML and can resolve subclonal heterogeneity.

## Genotyping of *BCR::ABL1* reveals more restricted leukemic phenotypes in ALL

Having observed somatic mutations in AML across myeloid, erythroid and megakaryocytic expression clusters, we investigated whether a similar distribution would also be detectable in acute lymphoblastic leukemia (ALL). We applied our approach to identify *BCR::ABL1* fusion genes in 4 cases of Philadelphia+ (Ph+) ALL. From ALL1 and ALL2, we obtained a bone marrow aspirate at initial diagnosis, while the other two samples were collected when ALL3 and ALL4 were in hematologic remission with measurable residual disease (MRD+). We readily detected *BCR::ABL1* transcripts that represented the p190 variant (better detected than the p210 variant commonly found in CML due to closer proximity of the fusion breakpoint to the 5′ end) (Fig. 6f). While no such fusion transcripts were detected at MRD+ remission, in ALL1 and ALL2, *BCR::ABL1*+ cells were detectable in two clearly defined clusters transcriptionally distinct from myelopoiesis, physiologic B

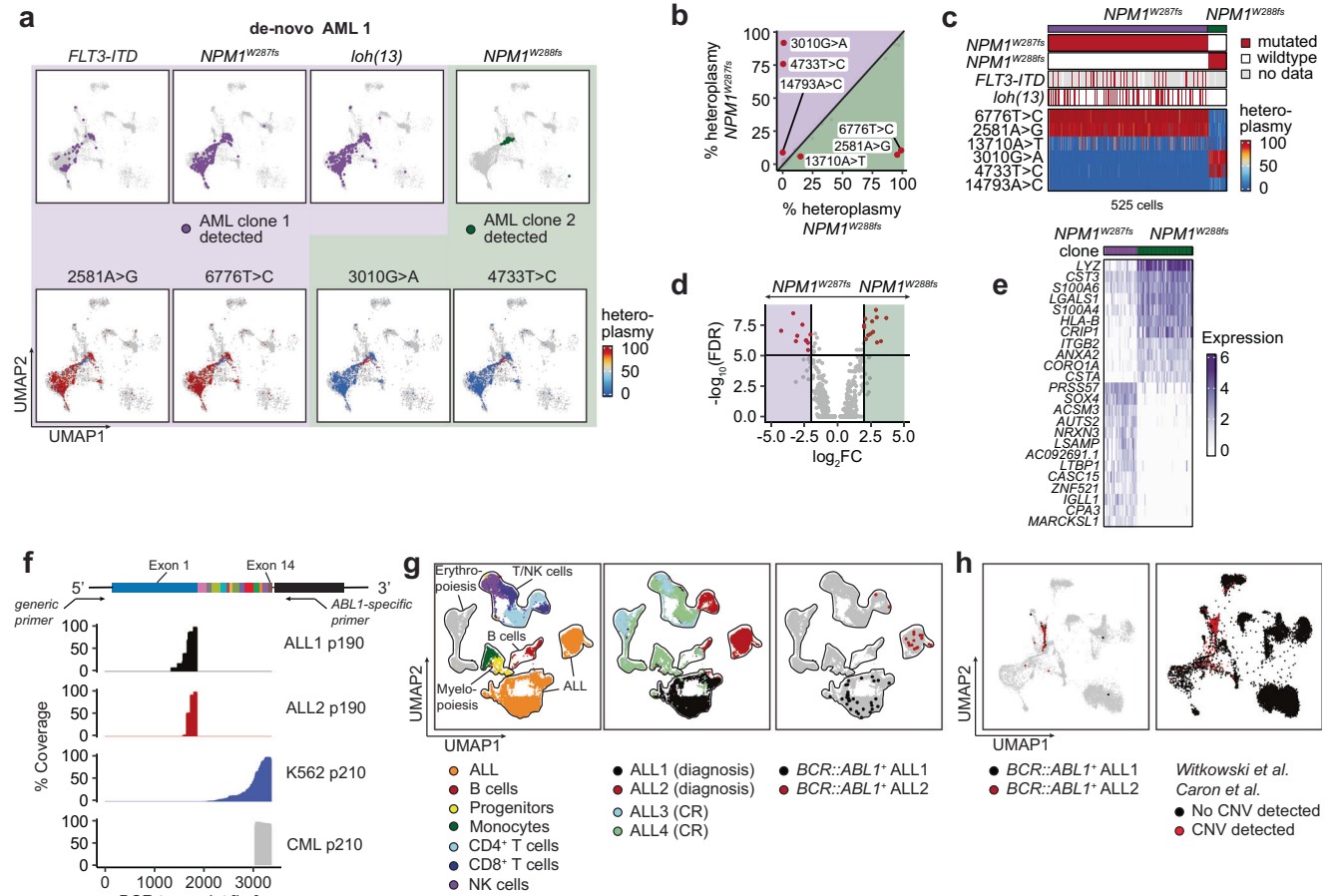

**Fig. 6 | Identification and tracking of leukemic cell populations in de-novo AML and Ph + ALL and CML. a** Co-existence of two subclones in de-novo AML 1. Detection of the somatic nuclear mutations *NPM1^{W287fs}* (clone 1) and *NPM1^{W288fs}* (clone 2) demonstrates co-existence of two AML clones that differ in the presence of *FLT3-ITD* and loss of heterozygosity on chromosome 13 (*loh(13)*) as well as several mitochondrial DNA mutations. **b** Differential analysis of bulk mitochondrial DNA heteroplasmy between clone 1 and clone 2 in de-novo AML 1. **c** Heatmap demonstrating molecular features of clone 1 and clone 2 in 525 cells of de-novo AML 1. **d**, **e** Differential gene expression analysis (DGEA) between GMP-like cells of clone 1 and clone 2. **f** Comparison of *BCR::ABL1* amplicons in two Philadelphia⁺ (Ph⁺) acute

lymphoblastic leukemia (ALL) cases, in K562 cells and Ph⁺ chronic myeloid leukemia (CML). In all 4 cases the same *ABL1*-specific primer was used. The p190 variant in ALL produces a shorter fusion transcript than the p210 variant in *CML* making it more detectable with targeted long-read sequencing from 10x Genomics cDNA libraries. **g** Identification of *BCR::ABL1⁺* cells in ALL bone marrow. UMAP plots show cell type annotation (left), samples (middle) and detection of *BCR::ABL1* transcripts (right) in ALL1 (black) and ALL2 (red). **h** *BCR::ABL1⁺* cells in bone marrow of ALL1 and ALL2 (left) and cells with detectable CNV changes in re-analyzed ALL datasets from refs. 33 and 34 (right), mapped to a healthy bone marrow reference.

cells or erythropoiesis, and thus consistent with B cell progenitor populations (i.e., upregulation of *CD34* or *DNTT* [encoding terminal deoxynucleotidyl transferase]) (Fig. 6g, Supplementary Fig. 12c-d). Annotation using a healthy bone marrow reference similarly mapped *BCR::ABL1⁺* cells to B cell progenitor-like cells (Fig. 6h-left). To confirm the constricted, predominantly B cell progenitor-like transcriptional profile of ALL, we characterized CNV changes in 48,188 single cell profiles of ALL bone marrow samples from two studies[33,34]. Indeed, the vast majority of cells harboring CNV changes (99%) had a B cell-like phenotype (Fig. 6h-right, Supplementary Fig. 12e), and these changes were mostly absent from other hematopoietic compartments. AML thus occupies a large phenotypic space, while ALL is far more confined.

**Tracking of genetic and transcriptomic variants in immune cells**
Our analysis of ETCTN 10026 AML bone marrow genotyping profiles also yielded identification of somatic mutations in co-localizing immune cell populations. In 4 of 9 AML cases, NK cells carried recurrent somatic mutations (Fig. 7a). In one notable example, NK cells harbored two *TP53* mutations (*TP53^{H179R}* and *TP53^{P278S}*), which were absent in AML (Fig. 7b), consistent with a clonal event in a progenitor population committed to NK cell differentiation. Re-analysis of bulk

amplicon sequencing data of the same patient at various timepoints revealed both *TP53* mutations to display an inverse trajectory compared to other somatic mutations (Fig. 7c). The resolution of single cell genotyping integrated with transcriptome-based phenotypes thus established the presence of mutations in non-leukemic cells, rather than what might have been otherwise interpreted as two competing AML subclones.

Our genotyping approach also improved detection of CD19⁺ CAR transcripts in scRNA-seq libraries, which is often incomplete[35]. This was accomplished by employing a *CD28*-specific primer to amplify CAR and wildtype *CD28* transcripts from a single cell cDNA library of an axicabtagene ciloleucel (Yescarta) infusion product (Fig. 7d). Our targeted detection of CAR transcripts identified not only almost all CAR⁺ T cells identified using unenriched Illumina scRNA-seq data but also found an additional 15% CAR⁺ T cells, corresponding to >75% CAR⁺ T cell detection (Fig. 7e, f, Supplementary Fig. 13a–c). Neither a primer closer to the 3′ end using a *CD247*-specific primer nor deeper sequencing coverage demonstrated by a downsampling analysis led to further increase in detected CAR⁺ T cells, once again due to the preferential amplification of shorter library fragments (Supplementary Fig. 13d, e). We noticed that the CD28 domain of axicabtagene

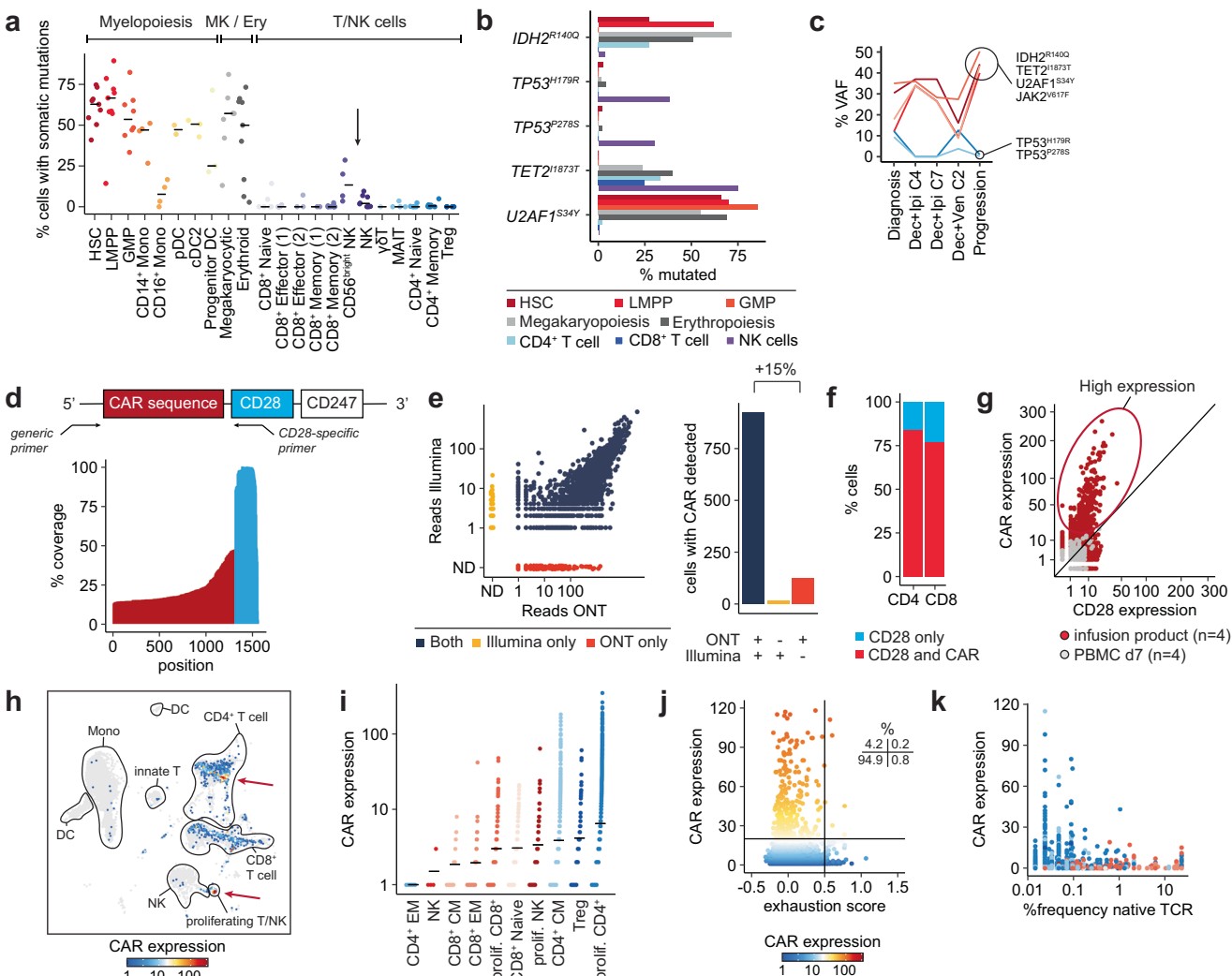

**Fig. 7 | Tracking of somatic mutations and CAR transcripts in T and NK cells.**
**a** Percentage of cells with detected somatic nuclear mutations across 22 hematopoietic cell types in 9 AML cases for a total of 17,911 cells. Arrow indicates NK cells. MK - megakaryopoiesis, Ery erythropoiesis, T/NK T and NK cells. **b** Percentage of cells with detected somatic nuclear mutations across hematopoietic cell compartments in AML1002. *TP53^R179H* and *TP53^P278S* are enriched in NK cells (purple). **c** Longitudinal clinical amplicon sequencing of bulk bone marrow-derived DNA demonstrates separation of NK-associated mutations (*TP53^R179H* and *TP53^P278S*) from AML-associated mutations in AML1002 (*IDH2^R140Q*, *TET2^I1873T*, *U2AF1^S34Y*, and *JAK2^V617F*). **d** PCR scheme demonstrating the amplification strategy of CAR transcripts with a CD28-specific primer. The coverage plot demonstrates the amplification of wildtype CD28 (blue) and fusion transcripts spanning into the CAR domain (red). **e** Read distribution with *nanoranger* on the Oxford Nanopore platform (ONT) compared to short-read sequencing (Illumina) as number of reads per cell barcode (left). The bar plot (right) shows the number of cells in which a CAR

transcript was detected with both technologies (+/+), only with Illumina (-/+) or only with ONT (+/-). ND not detected. **f** Quantification of cells that only have wildtype CD28 transcripts (blue, CD28 only) versus cells with at least one CAR transcript (red, CAR, and CD28) in one CAR T cell infusion product.
**g, h** Identification of CAR T cells with high expression of CAR transcripts by comparison with wildtype CD28 expression levels (**g**). Projection of CAR T cell detection on UMAP annotated by predicted cell types based on the PBMC reference dataset provided by Seurat (**h**). Data shown for four infusion products and four PBMC samples at day 7 after CAR T cell infusion. Red arrows indicate cells with high CAR expression. **i** Expression levels of CAR transcripts across ten predicted cell types in a total of 6504 cells. Expression levels of CAR transcripts compared to exhaustion score (calculated from expression of *PDCD1, CTLA4, TIGIT, HAVCR2, TOX, LAG3*, and *ENTPD1*) (**j**) and percentage clonal expansion of the native T cell receptor (TCR) relative to all detected TCRs within the library (**k**).

ciloleucel harbored a SNP that distinguishes wildtype *CD28* transcripts from *CD28* expression as part of the CAR transgene (Supplementary Fig. 13f, g), illustrating how germline SNPs can serve as proxies for molecular features in long-read sequencing data (i.e., "phasing").

When sequencing CAR transcripts using additional scRNA-seq libraries from CAR infusion products ($n = 4$) and matched blood samples 7 days after CAR infusion ($n = 4$), we repeatedly noticed cells with high levels of CAR transcript despite normal expression of wildtype *CD28*. These included proliferating CD4$^+$ and regulatory T cells (Tregs) (Fig. 7g–i; Supplementary Fig. 13h, i), consistent with the reported modulatory role of Tregs in CAR products[35]. To examine the origin and functional state of CAR high-expressing T cells, we used an established

gene signature of T cell exhaustion[18,24], and elucidated that high CAR expression was not predominantly associated with T cell exhaustion (4.2% non-exhausted vs. 0.2% exhausted) (Fig. 7j). Analysis of native TCR sequences indicated CAR-high expressing cells originate from previously unexpanded populations such as naïve T cells, consistent with polyclonal expansion of CAR T cell products (Fig. 7k)[36]. Together, CAR-high expressing cells from axicabtagene ciloleucel seemed to represent a subpopulation with high proliferation capacity without apparent transcriptional evidence of functional impairment.

Long-read sequencing of whole-transcriptome cDNA provided only shallow coverage of splice variants and thus was unsuitable for the interrogation of specific isoforms present in rare cell populations.

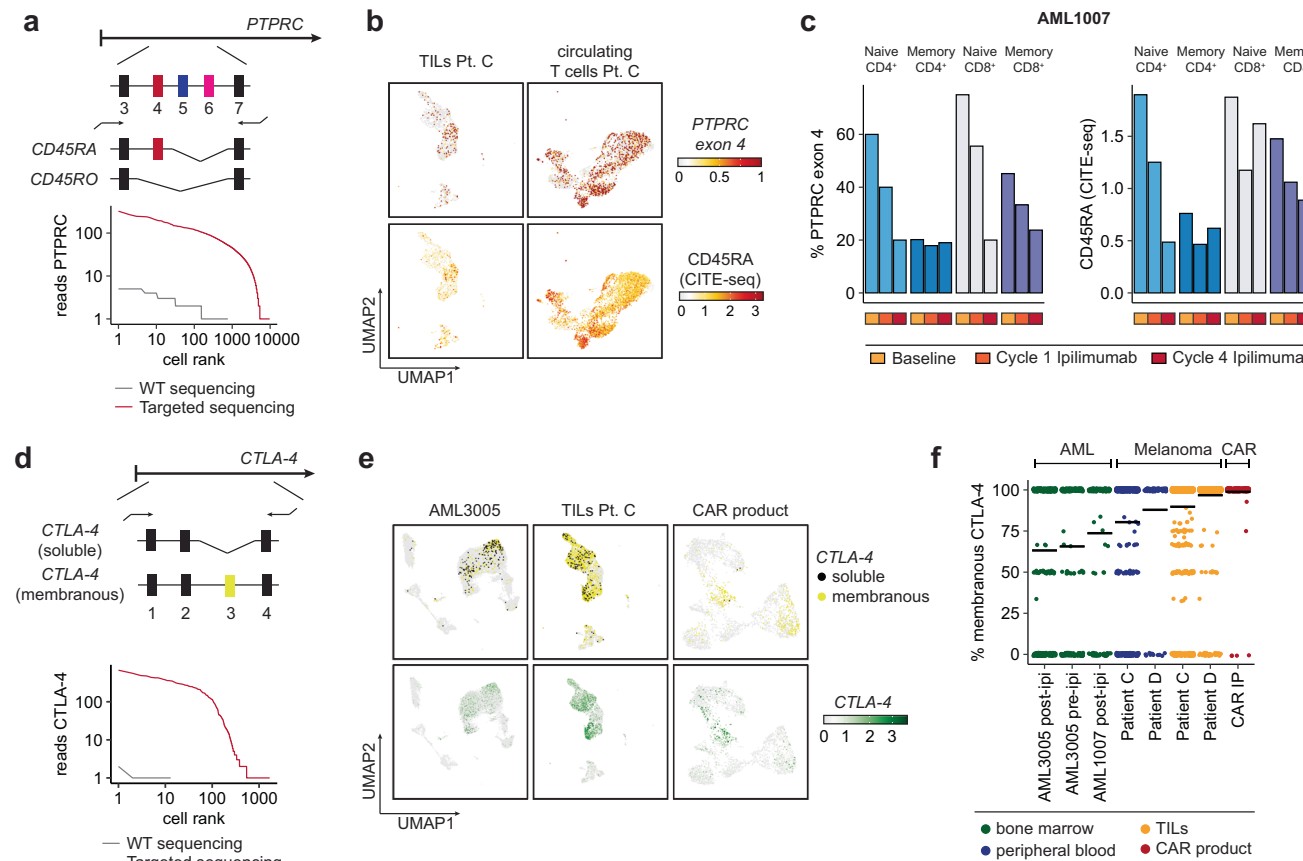

**Fig. 8 | Isoform detection in immune cells. a** Targeted amplification of *PTPRC* to detect differential splicing of exon 4 which determines expression of CD45RA (exon 4 expressed) versus CD45RO (exon 4 not expressed) (top). Targeted amplification dramatically increases coverage of *PTPRC* (red) compared to whole-transcriptome (WT) amplified cDNA (gray), both sequenced on the Oxford Nanopore platform. **b** UMAP representation of tumor-infiltrating T cells (TILs) and circulating T cells from melanoma Patient C (Oliveira et al., *Nature 2021*)[18]. The top row shows expression of *PTPRC* exon 4 (CD45RA) and the bottom row shows CD45RA protein expression measured by CITE-seq. **c** Expression of *PTPRC* (exon 4) (left) and CD45RA measured by CITE-seq (right) across T cell subsets in bone marrow of AML1007 before infusion of ipilimumab (baseline) and after 1 or 4 cycles of ipilimumab. **d** Targeted amplification of *CTLA-4* to detect exon 3 which discriminates the soluble (exon 3 absent) and membranous (exon 3 present) isoforms (top). Knee plot demonstrating the high degree of enrichment of *CTLA-4* transcripts with targeted long-read sequencing (red) versus whole-transcriptome long-read sequencing (gray) (bottom). **e** Expression of soluble (black) and membranous (yellow) isoforms of *CTLA-4* across AML3005, tumor-infiltrating T cells of Patient C, and a CAR T cell infusion product (top). Expression level of *CTLA-4* as measured by short-read sequencing indicates specific detection with the targeted approach (bottom). **f** Distribution of percentage of reads with membranous *CTLA-4* across eight different T cell single-cell cDNA libraries for a total of 4786 cells. Patients C and D were previously described by ref. [18]. CAR IP - CAR T cell infusion product.

Therefore, we used *nanoranger* to detect alternative splicing events. Reads that we used to detect somatic mutations in *U2AF1* also detected splice variants a, b and c of *U2AF1*. While the expression levels of variant a were highest and of variant c lowest, we observed no differences in the relative expression levels of the three isoforms in any of the major cell populations (Supplementary Fig. 14a). To track isoforms in T cells, we designed primers to target isoforms of *PTPRC*, *CTLA-4*, and *IL7R*. This approach dramatically increased the coverage of the *PTPRC* isoforms CD45RA and CD45RO, which are markers of naïve and memory T cells (Fig. 8a). To validate our workflow, we compared expression of *PTPRC* exon 4 and surface marker expression of CD45RA, which demonstrated reasonable concordance ($r = 0.45$) (Fig. 8b, Supplementary Fig. 14b). We used detection of *PTPRC* exon 4 in two cases of AML (AML1007, AML3005) following CTLA-4 blockade and observed reduced expression across T cell subsets, consistent with known T cell differentiation induced by ipilimumab (Fig. 8c, Supplementary Fig. 14c)[37].

Proteins encoded by isoforms of *CTLA-4* and *IL7R* exist in membranous and soluble forms, which differ in their function and are thus highly relevant for single cell studies of different immunological contexts. By targeting *CTLA-4* to dramatically increase the detection of its two isoforms, we observed that melanoma-infiltrating and CAR-T cells

had the highest expression level of the membranous variant, while circulating T cells in AML bone marrow and peripheral blood of melanoma patients expressed more soluble CTLA-4 (Fig. 8d–f). This could relate to sustained T cell activation in the tumor microenvironment and during the manufacturing process of CAR-T cell products. We noted that high expression of the *IL7R* isoform for the membranous variant was also detectable in CAR-T cells (Supplementary Fig. 14d).

## Discussion

Single cell RNA sequencing has substantially advanced our understanding of cell identities and fate. However, due to incomplete capture of long or lowly expressed transcripts and the inability to definitively detect many genetic and transcriptomic variants, current short-read single-cell data do not provide sufficient resolution to address many important biological questions that can be revealed through differential gene expression analyses of malignant versus non-malignant single cells or identification of lymphocyte subsets that are characterized by expression of different immune receptors and isoforms.

We overcame many of these limitations by designing a pipeline for the targeted detection of genetic and transcriptomic barcodes from single-cell cDNA libraries using long-read sequencing, and in turn, have

gained fresh insights into the technical and biological aspects of single cells recovered from a biopsy specimen. First, we addressed the challenge that current scRNA-seq genotyping assays often utilize large primer sets and would benefit from simplified workflows. We show that improvements in the accuracy of nanopore-based long-read sequencing, when coupled with scRNA-seq, afford the use of this platform for accurate genotyping of single nucleotide variants at single-cell resolution, an application that is highly dependent on low sequencing error. The ability to accurately sequence longer amplicons substantially streamlines workflows by reducing both the number of primers and processing steps required before loading a sample on a sequencer. Our use of long-read-based genotyping for unbiased tracking of AML clones and the identification of erythroid and mega-karyocytic cell populations that differentiate from leukemic clones demonstrates that a focus on myeloid progenitor populations for single cell genomic studies likely misses important AML subpopulations and suggests the need for a transcriptome-based classification of AML that includes erythroid and megakaryocytic signatures.

Second, we demonstrate that enriching for genes of interest augments the inherent ability of long-read sequencing to detect structural transcriptomic variants including alternative splicing events by increasing the number of cells with sufficient coverage for analysis by 2–3 orders of magnitude. This provides an opportunity to deeply interrogate cell populations for expression of specific molecular features such as individual isoforms, which provides qualitative information not obtainable by short-read data. For example, our *nanoranger* workflow makes it possible to characterize CD45RA expression when antibody detection via CITE-seq[38] proves inadequate. Current long-read flow cells generating 5–50 million reads enable primer panels designed to detect multiple informative isoforms to track immune cells or aberrant splicing events in cancer. Combining multiplex panels with barcoding approaches to lower per-sample sequencing costs enable systematic targeted identification in single cells. Further technical improvements to our approach could pave the way for its adoption as a clinical tool for immune monitoring in the future. Increasing the number of genotyped features per sample could extend target detection to other, less frequently mutated genes or even non-recurrent, private somatic mutations. This would make possible the detection and characterization of lowly abundant tumor cells, for example to identify the phenotype of residual malignant cells after systemic therapy or of relapse-initiating populations at incipient relapse. Such combined phenotypic and genetic information could potentially be used to instruct clinical decision-making and thus provides a clear translational application for our technology. Co-sequencing of DNA and RNA, currently under developed by several academic and industry groups[39–41], may overcome current limitations in genotyping while preserving the ability to dissect transcriptional states.

Lastly, long-read sequencing provided us with insights into the structure of 5′ 10x Genomics scRNA-seq cDNA libraries which are characterized by skewed coverage at the 5′ end and dramatic drop-off of coverage after the first 4kB. Therefore, we expect efforts for targeted enrichment of transcript regions will work best for shorter transcripts or loci within the first 4kB from the 5′ end. Besides potentially impacting gene expression quantification with short-read sequencing, this skewing is a major bottleneck for further developments in the field of single cell transcriptomics. While long-read sequencing can improve the detection of molecular features, the ceiling for any genotyping approach is currently determined by the low, incomplete representation of many transcripts including absence of longer variants and cannot be merely overcome through optimized amplicon-generation or deeper sequencing. We thus recommend future efforts to focus on developing high-throughput single cell chemistries that provide truly full-length cDNA in order to expand the accessible terrain in single cell RNA sequencing space.

## Methods

### Sample accrual and storage until analysis

AML bone marrow and peripheral blood samples were collected from study participants enrolled on the ETCTN/CTEP 10026 study (NCT02890329). Melanoma-infiltrating T cell were obtained from a participant in a phase I clinical trial at Dana-Farber Cancer Institute (DFCI) (NCT01970358)[18,42]. CAR T cells were obtained from a banked infusion product at DFCI[35]. ALL bone marrow samples were collected at DFCI. All study participants provided written consent and samples were collected under Institutional Review Board-approved protocols at DFCI. Until the time of analysis, bone marrow and peripheral blood samples were stored in vapor-phase liquid nitrogen after Ficoll-Hypaque density gradient centrifugation (Cytiva, Cat. no. 17144002) and cryopreservation with 10% dimethyl sulfoxide (Sigma-Aldrich, Cat. no. D2650).

### Processing of samples prior to single cell sequencing

Bone marrow and peripheral blood samples were slowly thawed in the vapor of a steam water bath at 37 °C. Thawing medium (phosphate buffered saline [PBS], Fisher Scientific, Cat. no. MT21040CV with 10% fetal calf serum [FCS], Gibco, Cat. no. 10437028 and 10% DNase I, grade II, Roche, Cat. no. 10104159001) was added in a drop-wise fashion until a total volume of 15 ml. After centrifugation for 5 min at 300 g, cells were resuspended in RPMI 1640 (Gibco, Cat. no. 11875119) with 10% FCS and 10% DNase I (Stemcell Technologies, Cat. no. 07900) and rested for 15 min at 37 °C. If cell viability assessed by 0.4% Trypan Blue solution (Sigma, Cat. no. T8154) was below 80%, dead cells were depleted (Dead Cell Removal Kit; Miltenyi, Cat. no. 130-090-101).

### Generation of single cell libraries

Single cell RNA sequencing libraries from AML bone marrow samples and CAR T cell infusion product were previously generated and their cDNA libraries were utilized for this work[24,35]. Single cell RNA sequencing libraries for the mixing experiment and from ALL bone marrow were generated for this study as follows: after resuspension at 1000 cells/μl in PBS with 0.04% ultrapure bovine serum albumin (BSA, Invitrogen, Cat. no. AM2616) 17,000 cells were loaded onto a Chromium Chip K (10x Genomics, Cat. no. 1000286). The Chromium Next GEM Single Cell 5′ Kit v2 (Cat. no. 1000263) was used for generation of single cell gene expression libraries. Enriched single cell TCR libraries were generated using the V(D)J Chromium Single Cell Human TCR Amplification Kit (Cat. no. 1000252). All library preparations were performed according to manufacturer's instructions. For short-read Illumina sequencing, libraries were pooled after quality control with a Bioanalyzer High Sensitivity DNA Kit (Agilent, Cat. no. 5067-4626). Sequencing was performed on an Illumina NovaSeq 6000 with 26/28 bp read1, 90 bp read2, 10 bp for index 1, and 10 bp index 2.

Single cell ATAC sequencing libraries were generated as previously reported[1,30]. During preparation of samples for sequencing, cells were subjected to fixation (formaldehyde) and permeabilization (NP-40 substitute) according to the mtscATAC-seq protocol[32]. After loading onto a Chromium Chip H (10x Genomics, Cat. no. PN-1000161) (targeted recovery of 7,000 cells), library preparation was performed with the Chromium Single Cell ATAC Library & Gel Bead Kit (Cat. no. PN-1000175) according to manufacturer's instructions. After quality control of libraries with a Bioanalyzer High Sensitivity DNA Kit (Agilent), pooled libraries were sequenced on a NovaSeq S2 platform (Illumina) with 50 bp paired-end reads, 8 bp for index 1 and 16 bp for index 2.

### Generation of amplicons for ONT sequencing

PCR reaction 1 (removal of template-switch oligo artifacts as previously described[15]):

5 μl of cDNA library was amplified under qPCR control until maximum exponential amplification (-5 cycles) with 2.5 μl 10 μM AA0272 and 2.5 μl 10 μM bio-AA0273 primers using 25 μl 2x KAPA HiFi Uracil+ Kit (Roche, Cat. no. 07959052001) and 5 μl 1:1000 nuclease-

free $H_2O$ pre-diluted 10,000x SYBR Green I Nucleic Acid Gel Stain (Invitrogen, Cat. no. S7563) in a total volume of 50 μl. 98 °C 3 min (initialization), cycles: 98 °C 20 sec (denaturation), 65 °C 30 sec (annealing), 72 °C 8 min (elongation), stop at end of elongation. After purification of PCR products with 65 μl ProNex Beads (1.3x) (Promega, Cat. no. NG2002) and elution in 20 μl, biotinylated fragments were captured using 5 μl 10 mg/ml Dynabeads (Invitrogen, Cat. no. 60101) according to manufacturer instructions and resuspended in 10 μl TE pH 8.0 (Invitrogen, Cat. no. AM9849).

**PCR reaction 2 (gene-specific amplification).** 1 μl resuspended bead-bound PCR 1 product was amplified using rhPCR[16] in a total volume of 50 μl under qPCR control until exponential amplification or a total of 18 cycles with stop at end of elongation. Primers are stored at −80 °C in TE pH 7.4 (Fisher Scientific, BP2476100). 1.25 μl 20 μM rhCGA_venus primer, 2.5 μl 5–20 μM patient-specific primer mix (for primers see Supplementary Tables 1–7), 2.5 μl 20x rhPCR buffer (300 mM Tris-HCl pH 8.4, 500 mM KCl, 80 mM $MgCl_2$), 0.8 ml 25 mM dNTPs (Fisher Scientific, Cat. no. FERR1121), 1.25 μl 20mU/μl RNase H2 (Integrated DNA Technologies, Cat. no. 11-02-12-01), 2 μl 5U/μl Hot Start One*Taq* DNA Polymerase (New England Biolabs, Cat. no. M0481L) and 5 μl 1:1000 nuclease-free $H_2O$ pre-diluted 10,000x SYBR Green I Nucleic Acid Gel Stain (Invitrogen, Cat. no. S7563). 95 °C 5 min (initialization), 96 °C 20 sec (denaturation), 60 °C 6 min (annealing), 72 °C 4 min (elongation). After purification of PCR products with 65 μl ProNex Beads (1.3x) (Promega, Cat. no. NG2002) and elution in 50 μl, biotinylated fragments were captured using 12.5 μl 10 mg/ml Dynabeads (Invitrogen, Cat. no. 60101) according to manufacturer instructions and resuspended in 10 μl TE pH 8.0 (Invitrogen, Cat. no. AM9849).

  20x rhPCR buffer:
- 3 mL 1 M Tris-HCl, pH 8.4 (300 mM) (Fisher Scientific, Cat. no. NC9922659)
- 5 mL 1 M KCl (500 mM) (Fisher Scientific, Cat. no. 50-842-959)
- 0.8 mL 1 M $MgCl_2$ (80 mM) (Fisher Scientific, Cat. no. 50-842-746)
- 1.2 mL nuclease-free $H_2O$ (Promega, Cat. no. MC1191)

**PCR reaction 3 (nested PCR).** 1 μl resuspended bead-bound PCR 2 product was amplified under qPCR control until end of exponential amplification with stop at end of elongation in a total volume of 50 μl. 25 μl Q5 High-Fidelity 2X Master Mix (New England Biolabs, Cat. no. M0492S), 5 μl 1:1,000 nuclease-free $H_2O$ pre-diluted 10,000x SYBR Green I Nucleic Acid Gel Stain (Invitrogen, Cat. no. S7563), 1.25 μl 20 μM CGAvenus.PS primer, 1.25 μl 20 μM each patient-specific nested primer mix. 98 °C 5 min (initialization), 98 °C 20 s (denaturation), 62 °C 30 sec (annealing), 72 °C 4 min (elongation). After purification of PCR products with 65 μl ProNex Beads (1.3x) (Promega, Cat. no. NG2002) and elution in 50 μl. Quantification of PCR products using TapeStation (Agilent, Cat. no. 5067–5365 and 5067–5366).

### Amplification of mitochondrial transcripts
Mitochondrial transcripts with high abundance were amplified only with PCR reactions 1 and 2 without streptavidin capture after PCR reaction 2. Mitochondrial transcripts with low abundance (*ND1*, *ND4*, *ND5*) were amplified using PCR reactions 1-3 as described above.

### Oxford Nanopore Technologies (ONT) sequencing
Sequencing adapters were ligated using the SQK-LSK114 kit (ONT) and the NEBNext Companion Module (New England Biolabs, Cat. no. E7180S) according to manufacturer instructions with 100 ng input per sample. Sequencing was performed using R10.4 flow cells on a MinION MK1C or GridION. Basecalling was performed using guppy version 6.1.2 with the module dna_r10.4_e8.1_sup. For optimization of sequencing costs, the Native Barcoding Kit 24 V14 (SQK-NBD114.24) was used to multiplex 2–10 amplicon libraries per R10 flow cell.

### Processing of ONT data using nanoranger
Following basecalling of long-read data with guppy[43], raw reads in the format of fastq files are processed by *nanoranger*. Required runtime parameters are a custom reference of transcripts for the enriched targets and a quantification mode which is either variant calling or VDJ reconstruction.

  Raw reads are aligned against the transcript reference using *minimap2*[44] with map-ont mode and --secondary=no option. Each sam record or "alignment" is subjected to a "deconcatenation" function using pysam[45]. In order to deconcatenate naturally or synthetically generated concatemers where each read can contain potentially many "sub-reads", primary and supplementary alignments are systematically processed. For each alignment two new fastq records are generated which mimic the read1-read2 format of standard paired-end libraries. The "read2" record is generated by saving a relevant part of the read based on the quantification mode. For variant calling the entirety of the "query" sequence is extracted. For VDJ reconstruction the query sequence (V gene transcripts used as reference in the input) and 100 nucleotides in the 3′ softclip of the alignment, which would cover the CDR3 region and J gene, are extracted. The "read1" record is generated by extracting the barcode region for each of the alignments. For 5′ 10x Genomics chemistry, the barcode region is assigned by search of a "motif" composed of the 10x forward primer, followed by 26 unknown bases representing barcode and UMI and the 10x TSO in a 200 nt window in the 5′ softclip of the alignment. This search is performed using the edlib package[46] within 5 Levenshtein distance under the decon_5p10XGEX function in the utils.py script. These parameters have been optimized heuristically to maximize the number of relevant candidate barcodes. To have more confident candidates the search window and edit distance can be decreased. To speed up the deconcatenation function input fastq files can be split into smaller parts and processed in parallel using –split flag as an optional input parameter.

  Following deconcatenation, barcode candidates are matched to a known whitelist of barcodes. To speed up the process while also allowing both mismatches and indels in candidate barcodes, the short-read aligner STAR[47] is used. First, a reference using whitelist barcodes is made after padding the barcodes with unknown (N) nucleotides on both sides to account for TSO and UMI nucleotides which are included in the candidate barcodes but are not to be penalized by the aligner based on errors on these parts. Next, the aligner is forced to align all bases of the candidates (as opposed to soft clipping the non-matching parts) to this reference by changing the alignment mode to EndtoEnd. Subsequently a sam file is generated where each candidate barcode will be matched with a clear start and end position to the best whitelist barcode. Only barcodes matching with a maximum of 1 mismatch or indel are considered. The UMI is extracted as the ten nucleotides that follow the last aligned base of the whitelist barcode on the read. Next, for variant calling mode, the "read2" fastq is aligned to the full genome using *minimap2* with splice mode. Barcode-UMI pairs are transferred to the final bam of primary alignments. The result of this process is an output similar to cellranger's possorted_genome_bam ready for downstream analysis. For the case of VDJ reconstruction, the "read2" fastq is processed using *MiXCR*[48]. Reads with assigned CDR3s are again paired with the matched barcodes. Code is available under https://github.com/mehdiborji/nanoranger.

### Variant calling
Functionality for variant calling is provided as an R package nanoranger.R (https://github.com/liviuspenter/nanoranger.R).

**Single nucleotide mutations including insertions/deletions.** For calling of single nucleotide variants (extract_mutation(), extract_indel() and extract_length_diff()), reads mapping to the locus were processed using pysam[45] by extracting the base for each read and any deletion or insertion using the *nanoranger* script perform_pileup.py. A filter for

the minimum number of reads per cell barcode and UMI is defined by the shape of the knee plot for each locus ranging from 5 to 100. Reads supporting a combination of cell barcode and UMI are collapsed by using starcode[49] to identify UMI clusters within a Levenshtein distance of 3. A consensus base and insertion/deletion is defined for each cell barcode and UMI combination based on the highest number of reads. A cell is considered to carry a mutation if >20% of UMIs support a single nucleotide variant or insertion/deletion.

**Fusion genes or CAR transcripts.** For analysis of fusion genes and CAR transcripts two strategies were utilized. For analysis of coverage across fusion genes, cell barcode, UMI, starting and end positions of reads mapping to the reference sequence of fusion genes (*BCR::ABL1* or *RUNX1::RUNX1T1*) or CAR transcripts were extracted with the *nanoranger* script fusion_gene.py using pysam[45]. Reads extending beyond the 5′ end of the fusion site were considered fusion reads. Reads extending into the CAR sequence when using a CD28-specific primer or extending into the CD28 sequence when using a CD247-specific primer are considered CAR reads.

Otherwise, reads were mapped to a reference containing individual genes (e.g., *BCR* and *ABL1*, *RUNX1* and *RUNX1T1* or CD28, CD247 and the full CAR sequence) followed by extraction of cell barcodes and UMI using fusion_gene.py. Reads were subsequently processed in analogy to single nucleotide mutations using extract_fusion().

**Mitochondrial DNA mutations.** Deconcatenated and genome aligned ONT reads were processed using maegatk[7] with the setting –NMmax 100. Mutations were called using the computeAFMutMatrix() function (https://github.com/petervangalen/MAESTER-2021) and further analyzed using custom scripts.

### Detection of isoforms

Differentially expressed exons were detected using the *nanoranger* workflow through a pileup approach for each exon in which the overlap of reads with the genomic coordinates of the exon were analyzed (isoforms.py) and quantified in R (extract_isoforms()). Absence of an exon was defined as less than 50% overlap with the genomic coordinates of the read. Alternatively, *nanoranger* was used with a reference genome and transcriptome containing individual isoforms of target genes.

### Gene expression analyses including annotation of cell types, donor/recipient origin and copy number changes

Raw Illumina sequencing reads were processed using CellRanger version 6.2.0 with the reference genome GRCh38-2020-A. Gene expression analysis was performed according to best practices with the Seurat package version 4.1.0[50]. Cell type annotation was performed either based on canonical marker gene expression or for AML samples by mapping single cell profiles to a healthy human bone marrow reference dataset provided by the Seurat package.

Donor and recipient annotation was performed by deconvoluting individuals with souporcell[25] using common SNPs and the parameter k = 2 followed by analysis of T cell and myeloid clusters taking into consideration clinical chimerism information.

Copy number changes were inferred using the numbat pipeline[51] considering structural chromosomal changes known from clinical karyotyping.

### Data reanalysis

For reanalysis of single cell RNA sequencing and genotyping data from van Galen et al. ref. 4, count matrices and genotyping data were downloaded from NCBI GEO (GSE116256). Gene expression data was processed using Seurat as described above. Genotyping information was integrated using a custom script.

Processed data from the beat AML[26] data set waves 1–4 was downloaded (http://www.vizome.org/) and processed using a custom

script. ALL scRNA-seq data were downloaded from NCBI GEO (GSE132509 and GSE130116) and processed using cellranger for quantification of count matrices and numbat for read-out of CNV changes.

### Comparison of *nanoranger* with genotyping of transcriptomes (GoT)

Bone marrow mononuclear cells from AML1022 were loaded onto two 10x lanes. One lane was processed according to the standard manufacturer's instructions and the second one according to the modifications of the 5′ GoT protocol described in the original publication (i.e., spike-in of gene-specific RT primers and additive primers during cDNA amplification) (Supplementary Tables 1–7)[5]. Both cDNAs were used as starting material for generation of *nanoranger* amplicon libraries. GoT cDNA served as input for generation of targeted amplicon libraries with an index PCR using locus-specific reverse primers. scRNA-seq libraries and the GoT amplicon library were sequenced together using the Illumina NextSeq 1000 system with a P2-100 cycle kit (read 1 26 cycles, read 2 90 cycles, index 1 10 cycles, index 2 10 cycles). ONT sequencing of GoT and *nanoranger* amplicon libraries was performed as described above.

### Primer design rhPCR primers

For the design of rhPCR primers the guidelines below were followed. The primers have a 6-nucleotide 3′ tail that begins with a ribonucleotide (rNNNNNM) containing a mismatched base (M) at position 6 followed by a spacer (/3SpC3/) to block polymerase extension.

For streptavidin capture, the 5′ end is conjugated with a biotin (/5Biosg/)

General design:

```
/5Biosg/NNNNNNNNNNNNNNNNNNNVrVNNNNM/3SpC3/
```

Rules for rhPCR primer design:
- $T_m$ 55 °C for sequence before 3′ 6-nt tail (all $T_m$'s from IDT OligoAnalyzer Tool with default settings)
- Primer length 18 to 24 nucleotides with 40-60% GC content for sequence before 3′ 6-nt tail
- No 4 G or C in a row
- No >5 A or T in a row
- 2-4 G or C in positions 1-5 of 3′ 6-nt tail
- No T before ribonucleotide or U as the ribonucleotide (indicated by V in the sequence above)
- For mismatch base use A, C or T (not G)
- T:G is not disruptive for mismatch base pair

### Primer design nested primers

The nested primers are positioned closer to the 5′ end of the transcript compared to rhPCR primers. An overlap of up to 5 nucleotides between rhPCR and nested primers is acceptable. The nested primers have a 5′ tail consisting of the mars sequence (AAGCAGTGGTATCAACGCAGAG).

General design:

```
AAGCAGTGGTATCAACGCAGAGNNNNNNNNNNNNNNNNNNNNNN
```

Rules for nested primer design:
- $T_m$ 57 °C
- Primer length 18 to 24 nucleotides with 40-60% GC content
- No G in last 2 3′ nucleotides
- Optimal 3′ end: AA, TA, CA, AT, AC, TT, CC
- GC-rich at 5′ end
- No 4 G or C in a row
- No >5 A or T in a row

### Benchmarking deconcatenation

Deconcatenation benchmarking was performed using *nanoranger* with the mode for analysis of whole-transcriptome cDNA generated with the 10x Chromium Next GEM Single Cell 3′ kit (v3.1),

concatemerized using MAS-Seq protocol and sequenced on PacBio Revio.

The publicly available circular consensus sequence data corresponding to run "DATA-Revio-PBMC-2" was downloaded from https://downloads.pacbcloud.com/public/dataset/MAS-Seq/. Following conversion into fastq files, *nanoranger* was run with the following modifications: instead of motif search for barcode-UMI candidates, the reverse of the Truseq Read 1 adapter (CTACACGACGCTCTTCCGATCT) was identified in the 200 nucleotides flanking the 3' end of each transcript alignment. In lieu of using only transcript references of the PCR targets, the entire gencode v44 transcriptome reference was utilized. This reference transcriptome was downloaded from https://ftp.ebi.ac.uk/pub/databases/gencode/Gencode_human/release_44/gencode.v44.transcripts.fa.gz.

Next, the number of extracted transcripts per 16-mer read was counted and compared to the number of segmented reads using the PacBio software for each array stored in the Sreads subfolder of the run DATA-Revio-PBMC-2 in the public data repository mentioned above. Subsequently, cell barcodes were corrected against the whitelist of the 10x Genomics 3' chemistry within one edit distance. Grouped reads for each cell barcode and deduplicated UMIs for each gene were summarized in a count matrix compatible with the output of *cellranger* for downstream purposes. As the count matrix provided by PacBio contained multiple entries for some genes, we collapsed these genes by summing up their UMI counts to enable a head-to-head comparison with *nanoranger*.

For our final analysis, we focused on shared genes between the PacBio and nanoranger count matrices, as the PacBio output missed a substantial amount of mitochondrial and ribosomal transcripts, likely due to differences in the underlying reference transcriptome.

### Benchmarking sequencing accuracy Oxford Nanopore Technologies (ONT) and Illumina

To perform sequencing error analysis using TCR reads obtained with ONT and Illumina, a dedicated alignment pipeline using the raw sequencing reads from V(D)J amplicon libraries generated from cDNA of T cells shown in Fig. 2 against a custom reference made of nucleotide sequences of the CDR3 regions was created.

For Illumina reads, read R2 was used, which was sequenced with 90 cycles. ONT reads were reprocessed using *nanoranger* by extracting a 120-nucleotide subsequence that confidently covered the CDR3 region. For the custom V(D)J reference, CDR3s with identical consensus nucleotide sequences in the *cellranger* and *nanoranger* output were selected and further filtered to have at least 50 supporting reads from *nanoranger* and 200 supporting reads from *cellranger*. We reasoned that ONT data had roughly half the sequencing depth and that each long read would most likely cover the CDR3 albeit with variable qualities, while short reads do not always cover the CDR3 (*cellranger* associates them with the CDR3 using shared UMIs with the reads that cover the CDR3). Since reads contained bases outside of the CDR3 regions, the consensus sequences were padded with "N" nucleotide on each side of the CDR3. STAR aligner was set to EndToEnd mode to ensure partially aligned reads covering CDR3 to be included in our analysis.

The number of mismatches and indels was extracted using AS and nM tags for each aligned read and reported per read and as average per CDR3.

### Amplification of GC-rich targets

For GC-rich targets, addition of 10% glycerol to the rhPCR (PCR2) and nested PCR (PCR3) markedly increased amplification and sequencing depth.

### Reporting summary

Further information on research design is available in the Nature Portfolio Reporting Summary linked to this article.

## Data availability

Gene expression matrices of samples sequenced for this work are deposited with the NCBI Gene Expression Omnibus (GEO)(accession number GSE243227). The raw long-read sequencing data generated for this work are available on the NCBI sequencing read archive (SRA) (project number PRJNA935418). Previously published gene expression matrices for samples from ETCTN/CTEP 10026 are available on NCBI GEO (accession GSE223844). Additional previously generated single cell RNA Illumina raw sequencing data are available on NCBI's Database of Genotypes and Phenotypes (dbGaP; https://www.ncbi.nlm.nih.gov/gap) under accession number phs003015.v1 (AML), or under accession number phs001451.v4.p1 (melanoma) and phs002922.v1.p1(CAR).

## Code availability

The code used to generate figures is available under https://github.com/liviuspenter/ONT-lineage-tracing [https://doi.org/10.5281/zenodo.10060863]. The nanoranger pipeline is available under https://github.com/mehdiborji/nanoranger and https://github.com/liviuspenter/nanoranger.R.

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

## Acknowledgements

We thank Doreen Hearsey and all staff from Ted and Eileen Pasquarello Tissue Bank in Hematologic Malignancies for excellent technical support with banking of clinical samples. We thank patients for their generous contribution of research samples for this study. Computationally intensive data processing was performed on the O$_2$ cluster of the Research Computing Group at Harvard Medical School. We thank Peter van Galen and Caleb Lareau for helpful discussions. This work was supported by National Institutes of Health, National Cancer Institute grant P01CA229092 (C.J.W.), UM1CA186709 (Principal Investigator: Geoffrey Shapiro), National Cancer Institute Cancer Therapy Evaluation Program, Bristol-Myers Squibb, LLS SCOR (7030-23), and LLS Therapy Accelerator Program. L.P. was supported by a research fellowship from the German Research Foundation (DFG, PE 3127/1-1) and is a Scholar of the American Society of Hematology, participant in the BIH Charité Digital Clinician Scientist Program funded by the DFG, the Charité – Universitätsmedizin Berlin, and the Berlin Institute of Health at Charité (BIH), is supported by the Max-Eder program of the German Cancer Aid (Deutsche Krebshilfe) and by funding from the Else Kröner-Fresenius-Stiftung (2023_EKEA.102). A.A. is supported by the Broad Institute IGNITE award. K.M. is supported by the ASCO YIA award. G.O. was supported by the Claudia Adams Barr Program for Innovative Cancer Research and by DF/HCC Kidney Cancer SPORE P50 CA101942. S.L. is supported by the National Institutes of Health, National Cancer Institute

Research Specialist Award (R50CA251956). J.S.G. is supported by the Conquer Cancer Foundation Career Development Award, Leukemia and Lymphoma Society Translational Research Program Award, and NIH K08CA245209. NCI CTEP provided study drug (Ipilimumab) support. This work was further supported by the CIMAC-CIDC Network. Scientific and financial support for the CIMAC-CIDC Network is provided through National Institutes of Health, National Cancer Institute Cooperative Agreements U24CA224319 (to the Icahn School of Medicine at Mount Sinai CIMAC), U24CA224331 (to the Dana-Farber Cancer Institute CIMAC), U24CA224285 (to the MD Anderson Cancer Center CIMAC), U24CA224309 (to the Stanford University CIMAC), and U24CA224316 (to the CIDC at Dana-Farber Cancer Institute). The CIMAC-CIDC website is found at https://cimac-network.org/. Panels in Fig. 1a, Fig. 2a, Fig. 3a, Fig. 5a, Fig. 5b, and Fig. 5d contain visual elements that were created with BioRender.com.

## Author contributions

L.P. designed and performed experiments, processed and analyzed data, and generated figures. M.B. developed the nanoranger processing pipeline and performed data analysis. A.N. designed and performed experiments. N.C., K.M., G.O., D.S.N. interpreted data. A.M.A., K.V.G. assisted with generation and processing of ISO-MAS-seq data. J.S.G., J.R., and R.J.S. provided clinical samples. H.L., W.S.L., and S.L. performed single-cell experiments. K.J.L. and C.J.W. designed and supervised the project. L.P. and C.J.W. wrote the manuscript.

## Competing interests

A.M.A. and K.V.G. are inventors on a licensed, pending international patent application, having serial number PCT/US2021/037226, filed by Broad Institute of MIT and Havard, Massachusetts General Hospital and Massachusetts Institute of Technology, directed to certain subject matter related to the MAS-seq method used in this manuscript. C.J.W. is an equity holder of BioNTech, Inc. and receives research funding from Pharmacyclics. D.N. received personal fees from Pharmacyclics, served as consultant to the American Society of Hematology Research Collaborative, and has stock ownership in Madrigal Pharmaceuticals. J.R. receives research funding from Kite/Gilead, Oncternal, and Novartis, serves on Scientific Advisory Boards for Akron Biotech, Clade Therapeutics, Garuda Therapeutics, LifeVault Bio, Novartis, Smart Immune, and TScan Therapeutics. J.S.G. reports serving on steering committee and receiving personal fees from AbbVie, Genentech, and Servier and institutional research funds from AbbVie, Genentech, Pfizer, and Astra-Zeneca. K.J.L. reports equity in Standard BioTools Inc. and serves on scientific advisory board for MBQ Pharma Inc. R.J.S. serves on the Board of Directors for Be the Match/National Marrow Donor Program and DSMB for BMS; reports personal fees from Vor Biopharma, Smart Immune, Daiichi Sankyo Inc, Neovii, Astellas and Jasper. The remaining authors declare no competing interests.
