## [Peer Review File · Nature Communications]

Integrative genotyping of cancer and immune phenotypes by long-read sequencingREVIEWER COMMENTS

Reviewer #1 (Remarks to the Author):

This article developed a long-read-based workflow and analysis pipeline (nanoranger) to detect cell lineage-defining features in single-cell cDNA libraries. The approach used limited primer sets to amplify target genes to detect various genetic barcodes in cell lineage through long-read sequencing. The authors demonstrated the effectiveness of this approach in tracking somatic and mtDNA mutations, fusion genes, transcriptomic structural variants, and the sequences of chimeric antigen and T cell receptors in cell lines. This method provides an efficient solution for detecting cell lineage mutations and supplements existing single-cell variant detection methods. This streamlined approach could advance the understanding of tumor and immune cell co-evolution at the single-cell level and have significant potential application value.

1. The authors may provide a more background introduction to the development of similar methods and discuss the strengths and weaknesses of those approaches.
2. Since accurate genotyping of single nucleotide variants at single-cell resolution is highly dependent on low sequencing error, the use of the PacBio platform seems to offer higher accuracy in transcriptomics. It would be valuable if the authors could further discuss the selection of sequencing platforms.
3. The authors may consider further discuss potential improvements to this method, such as expanding the detection of genetic variants to a larger set of genes or even to all genes at once.
4. The authors could discuss the potential applications of this method in scientific research as well as disease diagnosis and therapy.
5. In the sentence “We mixed Kasumi-1 (AML) with K562 (chronic myeloid leukemia [CML]) cells at four defined ratios ranging from 1:100 to 100:1 (Fig. 2d); at each ratio, we amplified and sequenced the homozygous TP53R248G mutation found in Kasumi-1 cells”, it may be helpful to improve understanding for reader if the authors clarify the differences between Kasumi-1 and K562 cells, as well as specify the exact four ratios used here.

Reviewer #2 (Remarks to the Author):

In this manuscript by Penter et al., the authors demonstrate the utility of the ONT platform for long-read sequencing and provide robust data to illustrate its effectiveness in detecting lineage-defining features, SNP variants, fusion genes, identifying CAR T-cells, and analyzing the TCR repertoire. This is a well-executed study; however, I have a few minor comments.

Minor comments:

Although the authors present sufficient data to support the applications of the ONT platform, several limitations have been identified, such as the low detection rate of specific fusion genes, the inability to identify certain mutation genotypes depending on their location, and allele dropouts due to low gene expression. I recommend that the authors briefly summarize these limitations in the abstract and discussion.

In the trajectory studies involving transplant-naïve AML/MDS patients and those in the post-HSCT setting, the authors report that erythropoiesis and megakaryopoiesis were entirely recipient-derived in 6 of the 8 analyzed cases post-HSCT. Please clarify whether this discordance was observed in the relapse setting or even before AML/MDS relapse, and provide information on the overall bulk donor chimerism of these patients at the time of the studies.

The observations made on erythroid and megakaryocytic progenitors may need to be tempered, as the number of genotyped cells is limited.

Regarding the finding of BCR-ABL restriction to B-cell progenitor-like cells, could the authors explain if this observation can be attributed to absence or very low expression of this transcript in other cell lineages?

In addition to identifying CAR T-cells and their functional status, it would be beneficial if the authors could also identify TCR transcripts to support the idea that high CAR-expressing cells represent a subpopulation with high proliferative capacity.

In the discussion, it would of interest to readers if the authors could compare the advantages and limitations of the ONT long-read platform with other available long-read sequencing platforms.

Reviewer #3 (Remarks to the Author):

Summary

This study introduces a repurposing technology of single cell cDNA technologies using priming of targets. The approach enables amplification of target regions that can then be sequenced with long-read technology which can genotype cells far more effectively than standard 10x sequencing. This has the major advantage of genotype-phenotype mapping at single cell resolution. The authors show how the technology can be applied in a wide ranging set of genotypic variation scenarios in the leukemia setting. Overall the paper represents an advance that will be of interest to the community. My comments are aimed at rounding out the paper such that while there is a laudable diversity of applications that are exemplified, the analytical components of how the methods are performing could be improved.

Critique:

1. There have been other methods attempting to recapture mutations from scRNA libraries. Could the authors show a more explicit comparison to, for example, PMID: 31270458 ? It would be most convincing to perform a head to head comparison on the same library with sensitivity/specificity metrics of known clonal mutations, or mutations that are lineage specific.
2. Pg 4 - this sentence ending with 'uncoupled from sample processing' is unclear to me. Please clarify or reword.
3. Pg 5 - the sentence 'Loci of interest...' please clarify in the intro a bit more detail of how the method is distinct from others referred to in this sentence : 'sophisticated primer panels are required to read out mutations using short-read sequencing, creating cumbersome and

inefficient experimental workflows'

4. A more rigorous presentation of the deconcatenation method is needed. Could the authors present a sensitivity/specificity analysis for the readers? This section reads as insufficiently quantitative to be convincing. More specifically, the reader is left without a sense of false negative and false positive rates. Moreover a discussion comment about how errors in deconcatenation would propagate and lead to spurious inference is warranted.

5. Pg 7 - can the CDR3 sequences be used to estimate the sequencing error rate in ONT using this method? This would be a nice and accessible comparison of the illumina approach and would allow for 'calibration' of sequencing error.

6. Pg 8 - how much would the TP53 frameshift lead to non-sense mediated decay. Is it possible to read this out of the data?

7. Pg 9, section ending with 'Thus, we caution that while detection of somatic mutations was specific for leukemic clones, their absence is not sufficient for identification of wildtype cells.' How much can phasing of heterozygous polymorphisms help here? Are there any SNPs recovered in the data- can this calibrate allele dropout vs absence of mutation more quantitatively?

8. Pg 9 - the the compound heterozygous TP53 mutations should be out of phase. Can you quantify this? Are the reads long enough such that both mutations are covered? Demonstrating they are out of phase would be very convincing.

9. Pg 12 - Phasing of SNPs might also help in deconvoluting donor/recipient cells- can the authors corroborate/support the mtDNA results with SNPs/haplotype analysis?

10. Discussion - curiously the discussion does not align well with the claims presented in the results. Suggest a rewrite of the discussion that more closely aligns with the main claims of the paper and their implications for the field. This is more of a high level subjective comment in that I found after reading the results, the discussion seemed disconnected.

11. Some approaches are now emerging that co-register DNaseq and RNAseq from the same cells. Could the authors discuss these emergent technologies - especially since there are potential issues of trying to genotype RNA : e.g. RNA edits, allele-specific expression due to epigenetic control, very low expressed transcripts etc.. that would yield incomplete or worse misinterpretation of variations. Although <https://pubmed.ncbi.nlm.nih.gov/36798358/> is not yet published, it is nevertheless highly relevant and should be placed in context.

Reviewer #4 (Remarks to the Author):

Penter et al. reported an integrative single-cell analysis method for simultaneous genotyping and phenotyping with the aid of long-read sequencing. They presented the utility of the platform by analyzing acute myeloid leukemia samples carrying several fusion genes.

Essentially this method is the combination of short-read and long-read error-correct sequencing of scRNA-seq libraries of the same origin. Using the long-read information, identification of SNVs (either on the chromosomes or mitochondrial DNA) or cell-specific sequences (such as TCR/BCR and CAR-T gene sequences) was possible as shown using cell lines and clinical samples. Discrimination between transcript variants was also possible and may help analyzing CD45RA/RO or CTLA-4 expression.

The drawbacks of the method are related to those of detecting mutations using cDNA. As the authors have shown using Kasui-1 and K562 cells, truncating variants are prone to nonsense-mediated decay and are difficult to identify from cDNA. Fusion genes having breakpoints that are >2,000 bp distant from the 5'- end are also difficult to identify because of incomplete reverse transcription.

While the reviewer believes in the importance of validating new systems, the most part of the manuscript (corresponding to figures 2-6) describes validation experiments using the method, and the presentation of results are superficial and are within the range of the knowledge of previous publications. The reviewer is unsure why the authors presented

fusion gene detection, mutation detection, mitochondrial profiling, and transcript variant analysis separately when the major strength of the system is to evaluate them simultaneously.

The reviewer is unsure why the authors analyzed ~1,000 cells per analysis when discussing the intratumor heterogeneity.

Reviewer #1

This article developed a long-read-based workflow and analysis pipeline (nanoranger) to detect cell lineage-defining features in single-cell cDNA libraries. The approach used limited primer sets to amplify target genes to detect various genetic barcodes in cell lineage through long-read sequencing. The authors demonstrated the effectiveness of this approach in tracking somatic and mtDNA mutations, fusion genes, transcriptomic structural variants, and the sequences of chimeric antigen and T cell receptors in cell lines. This method provides an efficient solution for detecting cell lineage mutations and supplements existing single-cell variant detection methods. This streamlined approach could advance the understanding of tumor and immune cell co-evolution at the single-cell level and have significant potential application value.

We thank the reviewer for their encouraging assessment of our work.

1. The authors may provide a more background introduction to the development of similar methods and discuss the strengths and weaknesses of those approaches.

We thank the reviewer for this great suggestion, which we have implemented accordingly. The introduction now goes into greater detail of strengths and weakness of similar methods:

Existing single-cell genotyping approaches include plate-⁴ and droplet-based protocols⁵⁻⁸, some of which entail adding locus-specific primers of predefined targets at the very initial steps of sample processing, at the stage of oil encapsulation. As all these approaches are based on Illumina short-read sequencing, they require large numbers of primers for covering all possible mutational sites across entire genes or genetic regions like the mitochondrial chromosome⁷, creating cumbersome and inefficient experimental workflows. While Illumina currently provides the lowest cost per base at sequencing error rates below 1/1000, a clear limitation of short-read based sequencing is that it is suboptimal for long fragments, which creates considerable difficulties for detecting mutation sites that require amplicons exceeding a length of 500 nucleotides⁹. Additionally, short-read sequencing is unable to easily resolve structural transcriptomic variants such as gene fusions, transgenes or even isoforms that characterize malignant and immune cell subpopulations. Finally, single-cell analyses often reveal unanticipated genetic and transcriptomic variants whose read-out would improve analytical resolution, thus creating the need to iteratively interrogate single-cell cDNA for such features even following library preparation.

2. Since accurate genotyping of single nucleotide variants at single-cell resolution is highly dependent on low sequencing error, the use of the PacBio platform seems to offer higher accuracy in transcriptomics. It would be valuable if the authors could further discuss the selection of sequencing platforms.

The reviewer is correct that low sequencing error is crucial for identification of single nucleotide variants and therefore the PacBio platform is a great choice. However, sequencing accuracy of Oxford Nanopore Technologies has also greatly improved recently and is considerably more affordable than the PacBio platform. We have added these points to our introduction:

PacBio and Oxford Nanopore Technologies (ONT) are two established long-read sequencing platforms. PacBio was the first long-read sequencing technology to achieve low error rates comparable to Illumina¹². However, with the recent introduction of the V14 chemistry, sequencing accuracy of ONT now exceeds 99%,¹³ while also having lower sequencing costs and providing a wider range of available flow cell sizes.

3. The authors may consider further discuss potential improvements to this method, such as expanding the detection of genetic variants to a larger set of genes or even to all genes at once.

Please see next point.

4. The authors could discuss the potential applications of this method in scientific research as well as disease diagnosis and therapy.

Points 3 and 4 are fantastic suggestions, which we have now adopted in our discussion:

Further technical improvements to our approach could pave the way for its adoption as a clinical tool for immune monitoring in the future. Increasing the number of genotyped features per sample could extend target detection to other, less frequently mutated genes or even non-recurrent, private somatic mutations. This would make possible the detection and characterization of lowly abundant tumor cells, for example to identify the phenotype of residual malignant cells after systemic therapy or of relapse-initiating populations at incipient relapse. Such combined phenotypic and genetic information could potentially be used to instruct clinical decision making and thus provides a clear translational application for our technology. Co-sequencing of DNA and RNA, currently under developed by several academic and industry groups.³⁹⁻⁴¹, may overcome current limitations in genotyping while preserving the ability to dissect transcriptional states.

5. In the sentence “We mixed Kasumi-1 (AML) with K562 (chronic myeloid leukemia [CML]) cells at four defined ratios ranging from 1:100 to 100:1 (Fig. 2d); at each ratio, we amplified and sequenced the homozygous TP53R248G mutation found in Kasumi-1 cells”, it may be helpful to improve understanding for reader if the authors clarify the differences between Kasumi-1 and K562 cells, as well as specify the exact four ratios used here.

We apologize for our lack of detail, which we have corrected in the revised version of our manuscript:

We mixed Kasumi-1 (an AML line harboring AML1::ETO and homozygous TP53^{R248G}) with K562 (a chronic myeloid leukemia [CML] line harboring BCR::ABL1 and monoallelic TP53^{Q136fs}) cells at four defined ratios (1:100, 1.5:1, 15:1, 100:1) (Fig. 2d); at each ratio, we amplified and sequenced the homozygous TP53^{R248G} mutation found in Kasumi-1 cells¹⁸.

Reviewer #2

In this manuscript by Penter et al., the authors demonstrate the utility of the ONT platform for long-read sequencing and provide robust data to illustrate its effectiveness in detecting lineage-defining features, SNP variants, fusion genes, identifying CAR T-cells, and analyzing the TCR repertoire. This is a well-executed study; however, I have a few minor comments.

We thank the reviewer for this concise summary and appreciate the opportunity to improve our work as suggested.

Minor comments:

Although the authors present sufficient data to support the applications of the ONT platform, several limitations have been identified, such as the low detection rate of specific fusion genes, the inability to identify certain mutation genotypes depending on their location, and allele dropouts due to low gene expression. I recommend that the authors briefly summarize these limitations in the abstract and discussion.

This is a great suggestion, which we have implemented accordingly. We have included the following sentence in the abstract:

Through systematic analysis of these classes of molecular features, we define the optimal targets for nanorange, namely those loci close to the 5' end of highly expressed genes with transcript lengths shorter than 4 kB.

These limitations are also addressed in the discussion, which we have further clarified:

Lastly, long-read sequencing provided us with insights into the structure of 5' 10x Genomics scRNA-seq cDNA libraries which are characterized by skewed coverage at the 5' end and dramatic drop-off of coverage after the first 4kB. Therefore, we expect efforts for targeted enrichment of transcript regions will work best for shorter transcripts or loci within the first 4kB from the 5' end. Besides potentially impacting gene expression quantification with short-read sequencing, this skewing is a major bottleneck for further developments in the field of single cell transcriptomics. While long-read sequencing can improve the detection of molecular features, the ceiling for any genotyping approach is currently determined by the low, incomplete representation of many transcripts including absence of longer variants and cannot be merely overcome through optimized amplicon-generation or deeper sequencing. We thus recommend future efforts to focus on developing high-throughput single cell chemistries that provide truly full-length cDNA in order to expand the accessible terrain in single cell RNA sequencing space.

In the trajectory studies involving transplant-naïve AML/MDS patients and those in the post-HSCT setting, the authors report that erythropoiesis and megakaryopoiesis were entirely recipient-derived in 6 of the 8 analyzed cases post-HSCT. Please clarify whether this discordance was observed in the relapse setting or even before AML/MDS relapse, and provide information on the overall bulk donor chimerism of these patients at the time of the studies.

We thank the reviewer for this query. The samples were indeed analyzed at time of overt AML/MDS relapse after transplant. We have included the information on non-fractionated bone marrow bulk donor chimerism in **Table 1** and mention the chimerism in the text.

Distinguishing normal and malignant hematopoiesis using expressed donor- and recipient-specific single nucleotide polymorphisms (SNPs) (souporecell)²⁴ revealed two additional AML-derived ECs, megakaryopoiesis and erythropoiesis, to be almost entirely recipient-derived in 6 of 8 analyzed cases from the post-HSCT cohort at time of relapse prior to initiation of decitabine and ipilimumab (non-fractionated bone marrow chimerism 2-85%) (Fig. 3b-left; Table. 1).

The observations made on erythroid and megakaryocytic progenitors may need to be tempered, as the number of genotyped cells is limited.

We agree with the reviewer that the number of genotyped cells is limited and that we have studied a subset of relapsed/refractory AML cases mostly in the setting of secondary AML. However, there are now two recent publications (Beneyto-Calabuig et al., Cell Stem Cell 2023 PMID 37098346 and Cores-Lopez et al., Cell Stem Cell PMID 37582363) that have also described erythroid and megakaryocytic differentiation states in AML, supporting our observations.

To address the reviewer's concern about the preliminary nature of these observations, we have added these points to the concluding statement of this section:

Altogether, our integrated analysis of recurrent somatic mutations, donor chimerism and copy number changes at single cell resolution revealed not only clear definition of individual AML clones present in myeloid cellular compartments but also their differentiation into erythroid and megakaryocytic lineage in the setting of relapsed/refractory secondary AML, consistent with two recent single-cell sequencing studies in AML/MDS^{8,29}. These discovery findings provide leukemia-discriminating signatures that can yield more accurate analysis of bulk RNA-sequencing profiles of AML.

Finally, we have now also added data on de-novo AML in which we genotyped several thousand cells that similarly demonstrate differentiation of AML into megakaryocytic and erythroid populations (new **Extended Data Figure 11b**, shown below).

Extended Data Fig. 11b. Percentage of mutated cells across myeloid, megakaryocytic and erythroid lineage in de-novo AML 1-3 assessed with somatic mutations.

Regarding the finding of BCR-ABL restriction to B-cell progenitor-like cells, could the authors explain if this observation can be attributed to absence or very low expression of this transcript in other cell lineages?

This is an important query. To address the question whether the absence of *BCR::ABL1* detection outside the ALL compartment could be attributed to low expression of the fusion transcript in other cell lineages,

we have included a new panel to **Extended Data Fig. 11d** in which we show the expression of the *BCR* transcript to be indeed largely constricted to the ALL compartment. As we are unable to perform additional DNA-based sequencing, it cannot be definitely ruled out using transcriptome-based genotyping with *nanorange*, whether this is the reason for the observed phenotypic restriction of *BCR::ABL1*⁺ cells. Nevertheless, this is the reason why we included a validation cohort in which we re-analyzed published scRNA-seq datasets by Witkowski et al. and Caron et al., in which we demonstrate that other copy number changes (i.e. other than t(9:22)) in ALL – especially in cases of Philadelphia⁺ ALL – also predominantly are detectable in progenitor B cells (**Extended Data Fig. 11e**). We believe this to be a strong confirmation of the genotyping results obtained with genotyping of *BCR::ABL1* using *nanorange*.

Extended Data Figure 11d-e. Genotyping of B ALL cells.

d UMAP plots show cell type annotation (left) and detection of *BCR* transcripts ALL1 and ALL2. **e** Detection of CNV changes in re-analyzed data of B ALL single cell profiles by Witkowski et al., Cancer Cell 2020⁴ and Caron et al., Scientific Reports 2020⁵.

In addition to identifying CAR T-cells and their functional status, it would be beneficial if the authors could also identify TCR transcripts to support the idea that high CAR-expressing cells represent a subpopulation with high proliferative capacity.

The reviewer proposes an attractive suggestion for further analyses of the single cell data by bringing together detection of CAR expression and TCR sequences. In fact, in the original submission, we already performed a comparison of CAR expression versus clonal expansion of the native TCR, which we calculated as the frequency of that TCR amongst all detected TCRs in the sample which was presented in **Fig. 7k**, and which we shown below. We observed that T cells with less expanded TCRs (i.e. naïve T cells) tended to have higher CAR expression, which supports the idea that naïve T cells are likely better suited for generating CAR T cell products.

Expression levels of CAR transcripts compared clonal expansion of native T cell receptor (**Fig. 7k**).

In the discussion, it would of interest to readers if the authors could compare the advantages and limitations of the ONT long-read platform with other available long-read sequencing platforms.

We agree with the reviewer that this is an important point which was also raised by reviewer 1. We have included a short discussion on the rationale of choosing ONT over PacBio in the introduction.

PacBio and Oxford Nanopore Technologies (ONT) are two established long-read sequencing platforms. PacBio was the first long-read sequencing technology to achieve low error rates comparable to Illumina¹². However, with the recent introduction of the V14 chemistry, sequencing accuracy of ONT now also exceeds 99%,¹³ while also having lower sequencing costs and providing a wider range of available flow cell sizes.

Reviewer #3

Summary

This study introduces a repurposing technology of single cell cDNA technologies using priming of targets. The approach enables amplification of target regions that can then be sequenced with long-read technology which can genotype cells far more effectively than standard 10x sequencing. This has the major advantage of genotype-phenotype mapping at single cell resolution. The authors show how the technology can be applied in a wide-ranging set of genotypic variation scenarios in the leukemia setting. Overall, the paper represents an advance that will be of interest to the community. My comments are aimed at rounding out the paper such that while there is a laudable diversity of applications that are exemplified, the analytical components of how the methods are performing could be improved.

We thank the reviewer for this summary and their favorable assessment of our work.

Critique:

1. There have been other methods attempting to recapture mutations from scRNA libraries. Could the authors show a more explicit comparison to, for example, PMID: 31270458? It would be most convincing to perform a head-to-head comparison on the same library with sensitivity/specificity metrics of known clonal mutations, or mutations that are lineage specific.

This is a terrific idea which we have implemented accordingly. Specifically, we have generated additional sequencing data with *nanoranger* in parallel with the genotyping of transcriptomes (GoT) protocol as suggested by the reviewer. As a result of this analysis, we have now added a new **Figure 3**, which we are also including below.

By targeting three somatic nuclear mutations in AML1022 (*DNMT3A*^{R882H}, *RUNX1*^{I177S}, *SF3B1*^{K700E}), we now show that *nanoranger* had a comparable performance to GoT for two of these targets. Notably, we are able to improve the performance of GoT by sequencing it with ONT and analyzing the respective data using *nanoranger*.

To benchmark performance of nanoranger to Illumina-based mutation detection, we processed a bone marrow sample of an AML case with three somatic mutations (DNMT3A^{R882H}, RUNX1^{I177S}, SF3B1^{K700E}) at relapse after allogeneic hematopoietic stem cell transplantation (HSCT) with nanoranger and with the 5' genotyping of transcriptomes (GoT) protocol⁵ (Fig. 3a). In addition to the Illumina sequencing described in the published protocol, we sequenced the unfragmented GoT library on ONT, processing the raw data with the nanoranger analytical pipeline. Overall, nanoranger and GoT sequenced with Illumina achieved similar genotyping rates (4.8-19.2% for nanoranger; 5.5-19.1% for GoT) of scRNA-seq profiles for two of the 3 mutations (DNMT3A^{R882H}, SF3B1^{K700E}) that we targeted (Fig. 3b-c). For RUNX1^{I177S}, located in close proximity to the 5' end, GoT reverse transcriptase (RT) primers improved capture from 15.1% to 39.2% (Extended Data Fig. 5a-c). For all three targets, we observed that the performance of GoT increased by another 1-7% when sequenced with ONT and processed with the nanoranger pipeline. Across all three experimental conditions, 99% cells with an identified SF3B1^{K700E} mutation were recipient-derived (Methods), demonstrating the specificity of these approaches (Fig. 3d). The largest difference between data acquired with nanoranger, GoT sequenced on Illumina and GoT sequenced on ONT was the cell barcode representation (Fig. 3e). We speculated that this could be in part due to a lower capture efficiency of longer library fragments with Illumina sequencing. We therefore analyzed the minimal fragment length in reads from the GoT library that associated with cell barcodes identified with Illumina and ONT sequencing versus those that were only identified using ONT sequencing. This revealed that Illumina sequencing did not capture fragments that were longer than approximately 1.5kB, demonstrating the advantage of long-read sequencing for genotyping of loci that are not immediately adjacent to the 3' or 5' of a transcript (Fig. 3f).

In sum, nanoranger has comparable performance to GoT, but genotypes different cell barcodes due to differences in sequencing capture rates of longer library fragments. The GoT ONT results with nanoranger processing indicate that including gene-specific RT primers during the cell

encapsulation step can improve the genotyping rate of targets close to the 5' end but requires the prescience to select targets prior to initiation of a single cell project. As illustrated by the numerous examples presented herein, the full nanoranger workflow enables re-analysis of archived cDNA libraries so that targets can be flexibly added to address new hypotheses that are generated after the initial single cell analysis.

Figure 3 Comparison of nanoranger with GoT.

a Experimental workflow of comparison between *nanoranger* and genotyping of transcriptomes (GoT). A pretreatment bone marrow sample of AML1022 at relapse after allogeneic hematopoietic stem cell transplantation (HSCT) was used for single cell cDNA library preparation according to the standard 10x Genomics 5' gene expression protocol and following the modified 5' GoT protocol with in-droplet inclusion of gene-specific reverse transcriptase primers. Both cDNAs were taken forward for sequencing with the standard *nanoranger* protocol (orange), GoT using Illumina sequencing (black) and GoT using Oxford Nanopore sequencing (blue).

b, c Number of cells genotyped with each experimental condition (**b**) and percentage of genotyped cells across hematopoietic differentiation states (**c**).

d Comparison of apparent single cell variant allele frequencies (VAFs) for SF3B1^{K700E} in donor- versus recipient-derived cells to demonstrate specificity of genotyping with each experimental condition.

e Comparison of cell barcodes identified with each condition. The venn diagrams demonstrate the number of cell barcodes that are uniquely identified or shared across experimental conditions. To enable direct comparison of captured cell barcodes, the cDNA for the GoT condition was used as input for *nanoranger*.

f Minimal read length versus number of reads for cell barcodes identified with GoT on Illumina and ONT (black) versus those identified only with GoT on ONT (blue), demonstrating the preferential sequencing of shorter fragments with Illumina sequencing.

2. Pg 4 - this sentence ending with 'uncoupled from sample processing' is unclear to me. Please clarify or reword.

We apologize for this unspecific wording and have clarified the sentence as follows:

*We report the generation of a long-read based pipeline using limited primer sets to amplify target genes from 5'-biased 10x Genomics scRNA-seq whole-transcriptome cDNA libraries, thus enabling the flexible detection of a wide range of barcodes from single cell libraries, **without spike-in of gene-specific primers during cDNA library preparation.***

3. Pg 5 - the sentence 'Loci of interest...' please clarify in the intro a bit more detail of how the method is distinct from others referred to in this sentence : 'sophisticated primer panels are required to read out mutations using short-read sequencing, creating cumbersome and inefficient experimental workflows'

We agree with the reviewer that the introduction was very short, which was also remarked by reviewer 1. We have expanded the introduction as follows:

Existing single-cell genotyping approaches include plate-⁴ and droplet-based protocols⁵⁻⁸, some of which entail adding locus-specific primers of predefined targets at the very initial steps of sample processing, at the stage of oil encapsulation. As all these approaches are based on Illumina short-read sequencing, they require large numbers of primers for covering all possible mutational sites across entire genes or genetic regions like the mitochondrial chromosome⁷, creating cumbersome and inefficient experimental workflows. While Illumina currently provides the lowest cost per base at sequencing error rates below 1/1000, a clear limitation of short-read based sequencing is that it is suboptimal for long fragments, which creates considerable difficulties for detecting mutation sites that require amplicons exceeding a length of 500 nucleotides⁹.

4. A more rigorous presentation of the deconcatenation method is needed. Could the authors present a sensitivity/specificity analysis for the readers? This section reads as insufficiently quantitative to be convincing. More specifically, the reader is left without a sense of false negative and false positive rates. Moreover, a discussion comment about how errors in deconcatenation would propagate and lead to spurious inference is warranted.

We thank the reviewer for this very valuable suggestion, which we have addressed in detail. We reasoned that by re-analyzing synthetically concatenated data that was originally sequenced and processed with PacBio and its standard tools, we would be able to demonstrate that *nanoranger* was able to sensitively and specifically deconcatenate such data at the level of individual reads, genes, and cells. We would like to point out that conceptually, *nanoranger* uses transcript alignment for identification of segments within reads. Therefore, segments that do not contain transcripts present in the reference transcriptome are ignored, while tools that are based on recognition of adapter sequences retain such segments.

*Finally, we assessed *nanoranger*'s performance for deconcatenating and quantifying MAS-ISO-seq data from healthy donor peripheral blood mononuclear cells (PBMCs) sequenced with PacBio. When comparing the number of segments deconcatenated per read, *nanoranger* found consistently fewer segments (median 12; range 0-37) than the PacBio processing tool *skera* (median 15; range 0-16), yielding a total of 85,387,903 and 110,127,015 segments (**Extended Data Fig. 1e-f**). This is because *nanoranger* identifies only segments that align to the reference transcriptome (gencode v44) and therefore does not recognize non-human transcripts, genomic contamination, intronic or unannotated transcripts, such as repeat elements. Nevertheless, the mean number of detected molecules per gene (141 vs. 55) and per cell (2,580 vs. 1,650) were highly correlated and consistently higher with *nanoranger* compared to *skera* ($r = 0.88$ and 0.97) (**Extended Data Fig. 1g-h**), likely due to differences in the underlying reference transcriptome and the annotation method. Similarly, *nanoranger* identified genes in more cells (122 vs. 50) and more genes per cell (1,380 vs. 867) ($r = 0.87$ and 0.96) (**Extended Data Fig. 1i-j**). This resulted in identification of very similar cell types between the two analytical pipelines but better capture of*

immunologically relevant genes such as HLA-E, IGHM or IL17RA with nanoranger (Extended Data Fig. 2a-b).

Benchmarking of nanoranger versus PacBio (Extended Data Fig. 1). Synthetically concatenated data sequenced on PacBio were processed with *nanoranger* and compared to results of the PacBio processing pipeline. **e, f** Number of segments extracted from each read with *nanoranger* and skera. **g, h** Number of identified molecules per gene (g) and per cell (h) with *nanoranger* and the PacBio analysis pipeline. **i, j** Number of cell barcodes associated with genes and number of genes detected per cell barcode with *nanoranger* and the PacBio analysis pipeline.

For Extended Data Fig. 2a-b see next page.

5. Pg 7 - can the CDR3 sequences be used to estimate the sequencing error rate in ONT using this method? This would be a nice and accessible comparison of the illumina approach and would allow for 'calibration' of sequencing error.

We thank the reviewer for this creative idea to utilize CDR3 sequences to determinate sequencing error rates of ONT compared to Illumina. We have implemented this approach as follows: based on consensus sequences that were determined for each TCR clone, we generated a TCR repertoire reference against which all TCR reads of that library were aligned. This permitted us to quantify the number of mismatches and indels of each read against the consensus.

This analysis demonstrated that we could validate the 99% sequencing accuracy claimed by Oxford Nanopore Technologies and demonstrates the ability of ONT to be used for detection of natural genetic barcodes. We show these analyses in the new **Extended Data Figure 2b-f**, also shown below).

The data also afforded us the ability to estimate the sequencing performance of V14 ONT chemistry and Illumina. By comparing TCR reads against their respective consensus sequence, we observed per-base mismatch (0.54% [Illumina] vs. 0.83% [ONT]) and indel rates (0.08% [Illumina] and 0.25% [ONT]) that were slightly higher with ONT ($p < 0.001$) (Extended Data Fig. 2c-e). Consistent with known characteristics of ONT^{10,18,19}, the indel rate increased in reads with homopolymers such as guanine-repeats (Extended Data Fig. 2f-g).

Comparison of sequencing error with Illumina and V14 Oxford Nanopore Technology (ONT) chemistry (Extended Data Fig. 2).

a, b UMAP projection of cell types identified based on count matrices generated with *nanoranger* and the PacBio analysis pipeline (**a**) alongside feature plots of genes with higher detection rate using *nanoranger* (**b**). **c** T cell receptor (TCR) reads were aligned against their respective consensus sequence, enabling to determine mismatch and indel rates of each read. **d** Per-base mismatch rate (left) and indel rate (right) of each read with Illumina and ONT sequencing. **e** Distribution of mismatches and indels per read for Illumina (grey) and ONT (orange). **f** Statistics of mismatch and indel rates with Illumina and ONT. **g** Rate of indels with Illumina (grey) and ONT (orange) increases with higher lengths of guanine homopolymers in TCR reads. Statistical testing with t-test.

6. Pg 8 - how much would the *TP53* frameshift lead to non-sense mediated decay. Is it possible to read this out of the data?

This is a fantastic idea. Assuming that two *TP53* alleles are each equally transcribed and one contains a frameshift mutation, it would indeed be possible to approximate nonsense mediated decay through a comparison of the wildtype and the mutated *TP53* transcript levels. However, K562 cells only have one mutated *TP53* allele, the other being deleted (PMID 17088437, 18277095, 8246608). This analytical approach is therefore not possible in K562. This is also reflected by the much lower expression of *TP53* in K562 that we show in Fig. 2f.

Figure 2f. Expression of *TP53* and percentage of cells with detectable *TP53* transcripts.

7. Pg 9, section ending with ‘Thus, we caution that while detection of somatic mutations was specific for leukemic clones, their absence is not sufficient for identification of wildtype cells.’ How much can phasing of heterozygous polymorphisms help here? Are there any SNPs recovered in the data- can this calibrate allele dropout vs absence of mutation more quantitatively?

Similar to the above query, the proposed strategy is in theory a very elegant way to overcome the issue of allelic drop-out. Through read-out of additional germline SNPs that colocalize on the same transcript from the same allele as the somatic mutation, it would be possible to distinguish true wildtype cells from those where the mutated allele was not captured. We have manually reviewed all amplicons that were generated for this manuscript, but unfortunately, we were unable to find a suitable SNP colocalizing on the same transcripts as a somatic mutation for this kind of analysis. This likely relates to the fact that on average germline SNPs are found only every 1-2kB (PMID 11237013) and most amplicons generated for our analyses have a length of only about 1kB or less.

Nevertheless, we did identify a germline SNP that distinguishes wildtype *CD28* from *CD28* as part of the expression vector of the CD19 CAR. To illustrate the feasibility of this approach, we now include additional analyses that demonstrate how a germline SNP can distinguish wildtype *CD28* from transgenic *CD28* (new **Extended Data Fig. 12f, g**, and also shown below).

Distinguishing wildtype *CD28* and *CD28* transcripts as part of the CAR construct using a single-nucleotide polymorphism (SNP) in *CD28* (Extended Data Fig. 12). **f** Length distribution of transcripts containing the *CD28* germline SNP (blue) versus the *CD28* SNP encoded by the CAR expression vector (red). **g** High correlation of identified CAR transcripts and transcripts containing the CAR-specific *CD28* SNP per cell.

*We noticed that the *CD28* domain of axicabtagene ciloleucel harbored a SNP that distinguishes wildtype *CD28* transcripts from *CD28* expression as part of the CAR transgene (Extended Data*

Fig. 12e-f), illustrating how germline SNPs can serve as proxies for molecular features in long-read sequencing data (i.e. “phasing”).

8. Pg 9 - the compound heterozygous *TP53* mutations should be out of phase. Can you quantify this? Are the reads long enough such that both mutations are covered? Demonstrating they are out of phase would be very convincing.

We thank the reviewer for this additional query regarding germline SNPs colocalizing to somatic mutations. As mentioned in response to query 6 and 7, K562 only express the *TP53* mutated allele, and do not have a wildtype *TP53* allele. Therefore, this analysis is not feasible using this cell line.

9. Pg 12 - Phasing of SNPs might also help in deconvoluting donor/recipient cells- can the authors corroborate/support the mtDNA results with SNPs/haplotype analysis?

We have indeed performed a comparison of mtDNA-based and SNP-based donor/recipient deconvolution (using the tool *souporcell*) as shown in **Extended Data Fig. 10g**. This demonstrates the high agreement of both methods for cells with sufficient coverage of mitochondrial transcripts.

		souporcell			
		recipient	donor	unassigned	
mtDNA	recipient	1,855	3	32	recipient
	donor	11	532	4	donor
	unassigned	519	1	36	unassigned

Extended Data Fig. 10g. Concordance of donor and recipient annotation with mtDNA variants and *souporcell*³. The relevant fields are highlighted. Due to insufficient coverage of cells with little mtDNA abundance, mtDNA variants are unable to assign some cells that *souporcell* can annotate.

10. Discussion - curiously the discussion does not align well with the claims presented in the results. Suggest a rewrite of the discussion that more closely aligns with the main claims of the paper and their implications for the field. This is more of a high level subjective comment in that I found after reading the results, the discussion seemed disconnected.

We thank the reviewer for this suggestion. In this revised manuscript we include a discussion that incorporates several more specific points that were raised by all four reviewers.

While long-read sequencing can improve the detection of molecular features, the ceiling for any genotyping approach is currently determined by the low, incomplete representation of many transcripts including absence of longer variants and cannot be merely overcome through optimized amplicon-generation or deeper sequencing. [...]

Further technical improvements to our approach could pave the way for its adoption as a clinical tool for immune monitoring in the future. Increasing the number of genotyped features per sample could extend target detection to other, less frequently mutated genes or even non-recurrent, private somatic mutations. This would make possible the detection and characterization of lowly abundant tumor cells, for example to identify the phenotype of residual malignant cells after systemic therapy or of relapse-initiating populations at incipient relapse. Such combined phenotypic and genetic information could potentially be used to instruct clinical decision making and thus provides a clear translational application for our technology.

Please also see the next comment.

11. Some approaches are now emerging that co-register DNaseq and RNAseq from the same cells. Could the authors discuss these emergent technologies - especially since there are potential issues of trying to genotype RNA : e.g. RNAedits, allele-specific expression due to epigenetic control, very low expressed transcripts etc.. that would yield incomplete or worse misinterpretation of variations. Although <https://pubmed.ncbi.nlm.nih.gov/36798358/> is not yet published, it is nevertheless highly relevant and should be placed in context.

This is a great suggestion. We now mention this work in our revised discussion.

Co-sequencing of DNA and RNA, currently under developed by several academic and industry groups.³⁹⁻⁴¹, may overcome current limitations in genotyping while preserving the ability to dissect transcriptional states.

Reviewer #4

Penter et al. reported an integrative single-cell analysis method for simultaneous genotyping and phenotyping with the aid of long-read sequencing. They presented the utility of the platform by analyzing acute myeloid leukemia samples carrying several fusion genes.

Essentially this method is the combination of short-read and long-read error-correct sequencing of scRNA-seq libraries of the same origin. Using the long-read information, identification of SNVs (either on the chromosomes or mitochondrial DNA) or cell-specific sequences (such as TCR/BCR and CAR-T gene sequences) was possible as shown using cell lines and clinical samples. Discrimination between transcript variants was also possible and may help analyzing CD45RA/RO or CTLA-4 expression.

We thank the reviewer for the summary of our work.

The drawbacks of the method are related to those of detecting mutations using cDNA. As the authors have shown using Kasumi-1 and K562 cells, truncating variants are prone to nonsense-mediated decay and are difficult to identify from cDNA. Fusion genes having breakpoints that are >2,000 bp distant from the 5'- end are also difficult to identify because of incomplete reverse transcription.

The reviewer is correct in this limitation. We would like to point out that this is a general issue of any transcriptome-based genotyping approach and not specific to our method. The strength of our approach is that we can identify multiple layers of information. By integrating different genetic barcodes with low and high genotyping rates (different somatic nuclear and mitochondrial DNA mutations), it is possible to greatly increase the number of cells for which at least one mutation can be detected.

While the reviewer believes in the importance of validating new systems, the most part of the manuscript (corresponding to figures 2-6) describes validation experiments using the method, and the presentation of results are superficial and are within the range of the knowledge of previous publications. The reviewer is unsure why the authors presented fusion gene detection, mutation detection, mitochondrial profiling, and transcript variant analysis separately when the major strength of the system is to evaluate them simultaneously.

We thank the reviewer for raising this question and agree with the reviewer that an integrated analysis of the different genetic features is a powerful way of utilizing our approach. However, there are few biological questions for which an integrated analysis of all possible genetic and transcriptomic features is useful, as their relevance is highly context specific. We would like to point out that our analyses already included several examples of integration of multiple data layers, such as CNV and somatic mutations (**Fig. 4**) or mitochondrial and somatic nuclear DNA mutations (**Fig. 5**).

Co-existence of two de-novo AML clones. Detection of the somatic nuclear mutations $NPM1^{W287fs}$ and $NPM1^{W288fs}$ demonstrates co-existence of two AML clones that are different in presence of $FLT3-ITD$ and loss of heterozygosity on chromosome 13 as well as several mitochondrial DNA mutations (**new Fig. 6c**)

Nonetheless, to further explore the ability for data integration, we now provide an additional analysis (new **Fig. 6c**, also shown above), in which we demonstrate the read-out of four different modalities: A 24-nucleotide internal tandem duplication ($FLT3-ITD$), two mutually exclusive $NPM1$ somatic mutations ($NPM1^{W287fs}$ and $NPM1^{W288fs}$), a loss of heterozygosity of chromosome 13 ($loh(13)$) and several mitochondrial DNA mutations. Together, they demonstrate a comprehensive characterization of an unusual case of two co-occurring AML clones that likely arose independently within the same patient.

This case illustrates the strength of our approach: currently available bulk sequencing methods likely would not have been able to identify these two co-occurring clones. Indeed, the clinical genotyping of this specimen only reported one of the two $NPM1$ mutations.

In another notable example (de-novo AML1) (Table 2), we discovered two leukemic clones defined by distinct mutually exclusive mutations in $NPM1$. Clone 1 ($NPM1^{W287fs}$) also harbored $FLT3-ITD$ and $loh(13)$, which were absent in clone 2 ($NPM1^{W288fs}$). By evaluating mtDNA mutations, we observed that both clones could be identified by a total of 6 mutually exclusive mtDNA mutations (Fig. 6a-c). Both clones were further distinguished by their phenotypes: clone 1 differentiated along the entire myeloid, megakaryocytic, and erythroid trajectory, while clone 2 had a more confined progenitor-like phenotype. These findings were confirmed by analyzing the distribution of the mtDNA mutations. Within GMP-like cells, further gene expression differences between both clones of de-novo AML1 could be identified such as differential expression of myeloid markers like LYZ or $CST3$ (Fig. 6d-e). This case demonstrates that, like secondary AML and MDS, de-novo AML can also differentiate from HSC-like to monocytic or even megakaryocytic and erythroid populations, which we also observed in two additional cases of de-novo AML (Extended Data Fig. 11a-b).

The reviewer is unsure why the authors analyzed ~1,000 cells per analysis when discussing the intratumor heterogeneity.

We thank the reviewer for this important query. The number of analyzed cells is dictated by the inherent features of cDNA that we discuss in our manuscript, like transcriptional bursting or the uneven coverage of transcripts from 5' to 3'. The numbers of genotyped cells that we were able to obtain are comparable to similar works that have used short-read sequencing for genotyping of AML single cell cDNA libraries (for example PMID 37098346 or 30827681).

Genotyping rate of different recurrently mutated genes in AML/MDS. Shown are the number of cell barcodes that are associated with high-quality cells (**Extended Data Fig. 6a**).

Due to these inherent limitations of current cDNA chemistries, the number of genotyped cells varies greatly for different targets. To illustrate this better, we provide an additional figure (new **Extended Data Fig. 6a**), in which we demonstrate that for genes such as splicing factors (*SRSF2*, *SF3B1* and *U2AF1*) or *NPM1*, which are all highly relevant for AML biology, the number of genotyped cells often exceeds 1,000. We anticipate that as the throughput of single cell technologies increases and new single cell chemistries optimized for long-read sequencing are introduced, the number of cells that can be genotyped will improve further.

REVIEWERS' COMMENTS

Reviewer #1 (Remarks to the Author):

The authors addressed all my comments.

Leng Han

Reviewer #2 (Remarks to the Author):

The authors have adequately addressed this reviewer's comments and concerns in this revised submission

Reviewer #3 (Remarks to the Author):

The authors have been very responsive to my comments. I have no further concerns and recommend publication.